

# Weakened impact of the Atlantic Niño on the future equatorial Atlantic and Guinean Coast rainfall

Koffi Worou[1], Hugues Goosse[1], Thierry Fichefet[1], and Fred Kusharski[2]

[1]Georges Lemaître Centre for Earth and Climate Research (TECLIM), Earth and Life Institute (ELI), Université catholique de Louvain (UCLouvain), Louvain-la-Neuve, Belgium
[2]Earth System Physics Section, Abdus Salam International Centre for Theoretical Physics, Trieste, Italy

**Correspondence:** Koffi Worou (koffi.worou@uclouvain.be)

**Abstract.** The Guinea Coast is the southern part of the West African continent. Its summer rainfall variability mostly occurs on interannual timescales and is highly influenced by the sea surface temperature (SST) variability in the eastern equatorial Atlantic, which is known as the Atlantic Niño (ATL3). Using historical simulations from 31 General Circulation Models (GCMs) participating in the sixth phase of the Coupled Model Intercomparison Project (CMIP6), we first show that these

models are able to simulate reasonably well the rainfall annual cycle in the Guinea Coast, with, however, a wet bias during boreal summer. This bias is associated with too high mean summer SSTs in the eastern equatorial and south Atlantic regions. Next, we analyze the near-term, mid-term and long-term changes of the Atlantic Niño mode relative to the present-day situation, in a climate with a high anthropogenic emission of greenhouse gases. We find a gradual decrease of the equatorial Atlantic SST anomalies associated with the Atlantic Niño in the three periods of the future. This result reflects a possible reduction

of the Atlantic Niño variability in the future due to a weakening of the Bjerkness feedback over the equatorial Atlantic. In a warmer climate, an oceanic extension of the Saharan Heat Low over the North Atlantic and an anomalous higher sea level pressure in the western equatorial Atlantic relative to the eastern equatorial Atlantic weaken the climatological trade winds over the equatorial Atlantic. As a result, the eastern equatorial Atlantic thermocline is deeper and responds less to Atlantic Niño events. Among the models that simulate a realistic rainfall pattern associated with ATL3 in the present-day climate, there

are 15 GCMs which project a decrease of the Guinean Coast rainfall response related to ATL3, and 9 GCMs which show no substantial change in the patterns associated with ATL3. In these 15 models, the zonal wind response to the ATL3 over the equatorial Atlantic is strongly attenuated in the future climate. Similar results are found when the analysis is focused on the rainfall response to ATL3 over the equatorial Atlantic. There is a higher confidence in the reduction of the rainfall associated with ATL3 over the Atlantic Ocean than over the Guinea Coast. We also found a decrease of the convection associated with

ATL3 in the majority of the models.

## 1   Introduction

The West African Monsoon (WAM) generally begins in mid-June and is characterized by a rapid shift of the rain band from the coastal areas to the Sahel region (Hansen, 2002; Sultan et al., 2005). As the rain band has moved northward, an upper level subsidence appears over the Guinea Coast. This causes the so-called "little dry season" in that area (Adejuwon and Odekunle,





2006; Fink et al., 2017; Wainwright et al., 2019), during which the rainfall decreases in magnitude compared to its mean over the April-May-June season. Over the 20[th] century, the interannual variability of the Guinean Coast rainfall was strongly influenced by the sea surface temperature (SST) fluctuations in the eastern equatorial Atlantic (Giannini, 2003; Polo et al., 2008; Suárez-Moreno et al., 2018). This oceanic area corresponds to the center of action of the leading oceanic mode of variability in the tropical Atlantic (Zebiak, 1993). Several terms are used to define this oceanic mode of variability: Atlantic equatorial

mode, Atlantic Niño and Atlantic zonal mode (Servain et al., 2000; Murtugudde et al., 2001; Ruiz-Barradas et al., 2000). In positive phases of the Atlantic Niño (hereafter referred to as ATL3), the eastern equatorial Atlantic is warmer than in a normal situation and the anomaly is amplified through the Bjerkness feedback (Bjerknes, 1969; Keenlyside and Latif, 2007; Lübbecke et al., 2018). The positive SST anomalies lower the surface pressure gradient over the equatorial Atlantic, and thus weaken the equatorial trade winds. This leads to westerly anomalies to the west of the abnormal warm surface area, which in turn

deepen the thermocline to the east. In consequence, the cooling due to the oceanic upwelling is reduced, which reinforces the initial warming. Moreover, another mechanism involving the southward advection of warm seawater from the subtropical North Atlantic can lead to the development of some ATL3 events (Richter et al., 2012). Other theories suggest an important contribution of the equatorial oceanic Rossby and Kelvin waves (Foltz and McPhaden, 2010a, b; Burmeister et al., 2016) and thermodynamic processes (Nnamchi et al., 2015) to the ATL3 variability. However, the Bjerkness feedback, i.e., the dynamical

forcing, has been shown to be the main driver of the Atlantic Niño variability (Jouanno et al., 2017).

The Atlantic Niño mode phases affect all the tropical band by modulating the Walker circulation. Moist air rises over the warm sea in the Atlantic basin and diverges in the upper levels, while in the Pacific and Indian Oceans, the flow converges and sinks at upper levels. By this Gill-type mechanism (Gill, 1980), an Atlantic Niño event can trigger a Pacific Niña (Wang,

2006; Losada et al., 2009; Rodríguez-Fonseca et al., 2009; Ding et al., 2011). Conversely, Tokinaga et al. (2019) found that the multi-year La Niña conditions could trigger a positive Atlantic Niño. During the monsoon season before 1970s, ATL3 positive phases favored an increase in the Guinean Coast rainfall, whereas the subsidence over the Sahel led to a decrease in rainfall over this region (Losada et al., 2010). This dipolar structure of the rainfall anomalous response to ATL3 over West Africa disappeared after 1970s. This feature was attributed to a strengthening of the covariability between SST in the eastern

tropical Atlantic and eastern equatorial Pacific (Losada et al., 2012). Destructive interferences between these two basins led to a non-stationary relationship between the Atlantic Niño and the rainfall in Sahel. By contrast, statistical analyses have demonstrated that the relation between SST changes in ATL3 region and the rainfall in the Guinea Coast during the boreal summer is stationary (Diatta and Fink, 2014; Rodríguez-Fonseca et al., 2015; Worou et al., 2020).

The new generation of General Circulation Models (GCMs) does show better skills in representing the atmospheric variables like temperature and precipitation over the globe (Tatebe et al., 2019; Zhai et al., 2020; Rivera and Arnould, 2020; Parsons et al., 2020; Almazroui et al., 2020; Akinsanola et al., 2020). In particular, results from these models show that from the fifth phase (CMIP5) to the sixth phase (CMIP6) of the Coupled Model Intercomparison Project, the surface temperature biases have been reduced over the tropical Atlantic, as pointed out by Richter and Tokinaga (2020) in an analysis of the pre-industrial con-



trol experiment performed with 33 models. Nevertheless, they found that in many models a warm bias remains in the east of the basin and along the western coast of Africa (Gulf of Benguela, western boundaries of West Africa), while cold biases exist in the west of the basin. Despite the presence of these biases, the models have improved their representation of the spatial patterns of the Atlantic Niño mode. Several years earlier, Kucharski and Joshi (2017) showed that the South Atlantic Ocean Dipole, which involves the SST variability in the ATL3 region and in the southwestern Atlantic, off the Argentina–Uruguay–Brazil

coast (Nnamchi and Li, 2011; Nnamchi et al., 2011), was already well represented in CMIP5 models and the new analyses confirm thus this good performance.

Tokinaga and Xie (2011) have revealed a weakening of the ATL3 variability during the last decades. They argued that this decrease over the period 1950 - 2009 was caused by the anthropogenic aerosol forcing, whose cooling effect in the northern

Atlantic was greater than in the southern Atlantic. As a result, the SST meridional gradient over the tropical Atlantic weakened and led to a relaxation of the equatorial trade winds and a deepened thermocline to the east. It was followed by a reduction of the upwelling of cold water from the deepest levels which reinforces the positive SST trend and reduces the ATL3 variability. On the other hand, the exchange of the turbulent latent heat flux between the surface and the atmosphere represents an important negative feedback in the growing of the SST anomalies during the ATL3 events. Prigent et al. (2020) attributed the

decreased variability of the ATL3 since 2000 to a weakening of the Bjerkness feedback in the equatorial Atlantic, as Tokinaga and Xie (2011), and also to an increased cooling of the sea surface due to an increased latent heat flux release to the atmosphere. These authors found a reduced sensitivity of winds in the western equatorial Atlantic basin to the ATL3, potentially due to a northward migration of the mean intertropical convergence zone (ITCZ), and a westward migration of the Walker circulation rising branch in the tropical Atlantic. In the eastern equatorial Atlantic, they found a reduced coupling between the surface and

the thermocline.

Rainfall changes over the Guinea Coast during the past decades follow the observed decrease of the ATL3 variability (Tokinaga and Xie, 2011; Worou et al., 2020), underlying the strength of the ATL3 impact on the rainfall activity in this area (Nnamchi et al., 2021). Thus, the question arises whether the Atlantic Niño will change in the future and what will the implica-

tions be for the tropical hydroclimate, particularly over the Guinea Coast. Mohino and Losada (2015) have shown an eastward displacement of the positive rainfall anomalies induced by ATL3 in the tropical Atlantic under a warmer climate. Another study has revealed a weakening of the relation between Atlantic Niño and El-Niño events in a warmer climate, due to a faster warming of the mid-troposphere relative to the low levels, leading to a more stabilized atmosphere and a reduced convection (Jia et al., 2019).

In the present study, we provide a more detailed analysis of the future changes in the Atlantic Niño and their impact on the rainfall over the tropical Atlantic and land masses in Guinea Coast by using GCM results obtained within CMIP6. The near-term, mid-term and long-term changes are analyzed separately. In section 2, we describe the data used in our analyses. Section 3 focuses on the annual cycle of the Guinea Coast rainfall and the SST in the ATL3 region in models and observations. In section 4, we evaluate the SST and rainfall patterns associated with the Atlantic Niño in the CMIP6 models over the last 30 years of the





historical simulations conducted with these models. We highlight the group of models that simulate a realistic rainfall pattern associated with ATL3 over the Guinea Coast. Section 5 discusses the modelled future changes of the Atlantic Niño and their impact on the rainfall over the tropical Atlantic and Guinea Coast, as well as the associated mechanisms. In the last section, we draw the main conclusions.

## 2   Data and methods

This study focusses on the July-August-September (JAS) season, during which the rainfall variability over the Guinea Coast and the entire West Africa peaks (Nnamchi et al., 2021). Outputs from 31 GCMs participating in CMIP6 are analyzed. Two kinds of simulations are considered: historical and high-emission scenario climate experiments. The historical climate simulations cover the 1850-2014 period, with realistic natural and anthropogenic forcings. The climate projections are based on the Shared Socioeconomic Pathway 5, with a global radiative anthropogenic forcing that reaches $8.5\,\mathrm{W}\cdot\mathrm{m}^{-2}$ at the end of the 21$^{\text{st}}$ century (SSP5-85). These latter simulations started in 2015 and assume a world with an intense industrial activity based on the fossil fuel and a large economic growth (O'Neill et al., 2016). Only one member of each model is considered for each period. The description of the data is available in Table 1.

The ERA5 dataset, a reanalysis product from the European Centre for Medium-Range Weather Forecasts (ECMWF, Hersbach et al., 2020), is used as a reference to evaluate the performance of the CMIP6 models in simulating the surface and atmospheric fields under present-day conditions. ERA5 has replaced ERA-Interim (Dee et al., 2011) that has been widely used to study the characteristics of the West African climate (Manzanas et al., 2014; Lavaysse et al., 2015; Kebe et al., 2016; Raj et al., 2018; Maranan et al., 2018; Diakhaté et al., 2019, 2020; Wainwright et al., 2019). ERA5 for instance shows an improvement of the rainfall representation over Burkina Faso (Tall et al., 2019), which is a part of the Sahelian region. We also use the monthly sea surface height (SSH) data from the Ocean ReAnalysis System 5 (ORAS5, ECMWF, Zuo et al., 2019). Finally, a set of monthly observational precipitation and SST data is compared with both the historical model outputs and reanalysis data. The observed rainfall data are the following:

- CPC Merged Analysis of Precipitation (CMAP) which is a combination of satellites and rain-gauge measurements, and NCEP-NCAR reanalysis outputs (Xie and Arkin, 1997);

- Global Precipitation Climatology Project (GPCP) which is also a merge of rainfall information from satellites and in-situ observations (Adler et al., 2016);

- Climate Research Unit Time Series (CRUTS4.03), a collection of rainfall data from more than 4000 weather stations (Harris et al., 2020).

Observed monthly SST data are derived from :

- the version 1.1 of the Hadley Centre Sea Ice and Sea Surface Temperature dataset (HadISST, Rayner et al., 2003);

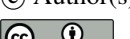



- the version 5 of the NOAA Extended Reconstructed SST (ERSST, Huang et al., 2017);

- the COBE-SST2 dataset (Hirahara et al., 2014).

The evaluation of the present-day climate simulations is based on the last 30 years of the historical climate experiments (1985-2014). Three different periods are considered for the future climate analyses: the near-term (2015-2039), mid-term (2040-2069)
and long-term (2070-2099) periods. Prior to any diagnostic over each period, the data are interpolated to the same grid of $1°$ resolution. Except for the analysis of the annual cycles and the seasonal means of the various surface and atmospheric fields, the monthly data anomalies at each grid point are computed and then quadratically detrended. The data are averaged over July, August and September to get the seasonal anomalies. Much of the statistics in this paper are based on linear regressions onto the standardized SST index of the Atlantic Niño. This index is defined as an average of SST anomalies over the ATL3 region
(Table 2, Zebiak, 1993).

The difference of the wind divergence between 200 hPa and 850 hPa levels (DIV200/850) is used to evaluate the direction and intensity of the vertical air motion (Gómara et al., 2017; Diakhaté et al., 2019). Positive values of this difference (upper level minus low level) means that there is a low level convergence of the wind, an ascent and a divergence at high level. The
Root Mean Square Error (RMSE) is also used to compare the deviation between each model and ERA5. The spread of the 31 GCMs is evaluated with the Interquartile range (IQR) metric. Finally, similar to Jia et al. (2019), a sign-dependent average of the rainfall regression coefficient is utilized to select the models based on their ability to simulate the rainfall pattern over the equatorial Atlantic and the Guinea Coast regions (Table 2). This technique is described as followed. Over each of these two regions, the rainfall regression coefficients that are significant at 95 % confidence level (two-sided Student test) are identified.
Their mean over the considered region gives a whole average coefficient. Then, over each region, the sign-dependent average is computed as the area average of the significant regression coefficients that have the sign of the whole average. The sign-dependent average is set to zero if there is no significant regression coefficient.

For a given period, when the regression patterns are averaged over some subsets of models, we determine the robustness
of the result by adapting the method of Tebaldi et al. (2011). The signal in areas where at least 50 % of the models show a significant regression (two-sided Student test at 95 % confidence level) and where at least 80 % of the models agree on the sign of the mean is considered as robust. When we consider the change between two periods, the sign of the change averaged over a subset of models is considered as robust if at least two thirds of the models agree on the multimodel mean (Rehfeld et al., 2020), and at least the change of the multimodel mean is significant at 95 % confidence level according to a two-sided Welch
t-test. Finally, coordinates of different geographic areas used in the present study are defined in Table 2.



**Table 1.** CMIP6 models and members of historical and SSP5-85 simulations. (*) indicates models for which the sea surface height variable was not yet available at the time of this study. Models for which, on the one hand, the sea level pressure and the 10 m horizontal wind variables, and, on the other, the mixed layer depth variable were not used for the mean state change analyses are indicated by (**) and (***), respectively. The other variables used are: rainfall, sea surface temperature, 850 hPa specific humidity, 850 hPa and 200 hPa horizontal wind components.

| CMIP6 model | Historical member | SSP5-85 member |
|---|---|---|
| ACCESS-CM2 | r1i1p1f1 | r1i1p1f1 |
| ACCESS-ESM1-5 | r1i1p1f1 | r1i1p1f1 |
| BCC-CSM2-MR | r1i1p1f1 | r1i1p1f1 |
| CAMS-CSM1-0 | r1i1p1f1 | r1i1p1f1 |
| CanESM5 | r1i1p1f1 | r1i1p1f1 |
| CanESM5-CanOE | r1i1p2f1 | r1i1p2f1 |
| CESM2 (*, **, ***) | r11i1p1f1 | r2i1p1f1 |
| CESM2-WACCM (**) | r1i1p1f1 | r1i1p1f1 |
| CNRM-CM6-1 | r1i1p1f2 | r1i1p1f2 |
| CNRM-CM6-1-HR | r1i1p1f2 | r1i1p1f2 |
| CNRM-ESM2-1 | r1i1p1f2 | r1i1p1f2 |
| EC-Earth3 | r1i1p1f1 | r1i1p1f1 |
| EC-Earth3-Veg | r1i1p1f1 | r1i1p1f1 |
| FGOALS-f3-L (*) | r1i1p1f1 | r1i1p1f1 |
| FIO-ESM-2-0 (**, ***) | r1i1p1f1 | r1i1p1f1 |
| GFDL-ESM4 (*, ***) | r1i1p1f1 | r1i1p1f1 |
| GISS-E2-1-G | r1i1p1f2 | r1i1p1f2 |
| HadGEM3-GC31-LL | r1i1p1f3 | r1i1p1f3 |
| INM-CM4-8 (***) | r1i1p1f1 | r1i1p1f1 |
| INM-CM5-0 (***) | r1i1p1f1 | r1i1p1f1 |
| IPSL-CM6A-LR | r1i1p1f1 | r1i1p1f1 |
| KACE-1-0-G (*, **, ***) | r1i1p1f1 | r2i1p1f1 |
| MCM-UA-1-0 (*,***) | r1i1p1f2 | r1i1p1f2 |
| MIROC6 (**) | r1i1p1f1 | r1i1p1f1 |
| MIROC-ES2L (***) | r1i1p1f2 | r1i1p1f2 |
| MPI-ESM1-2-HR | r1i1p1f1 | r1i1p1f1 |
| MPI-ESM1-2-LR | r1i1p1f1 | r1i1p1f1 |
| MRI-ESM2-0 | r1i1p1f1 | r1i1p1f1 |
| NorESM2-LM (**) | r1i1p1f1 | r1i1p1f1 |
| NorESM2-MM (**) | r1i1p1f1 | r1i1p1f1 |
| UKESM1-0-LL (**) | r1i1p1f2 | r1i1p1f2 |



**Table 2.** Coordinates of different domains.

| Domain | Lon min (°) | Lon max (°) | Lat min (°) | Lat max (°) |
|---|---|---|---|---|
| West Africa box (WAB) | −20 | 10 | 4 | 20 |
| Guinea Coast box (GCB) | −20 | 10 | 4 | 10 |
| ATL3 box (ATL3B) | −20 | 0 | −3 | 3 |
| Equatorial Atlantic box (EAB) | −30 | 10 | −5 | 5 |
| Tropical Atlantic box 1 (TAB1) | −70 | 10 | −20 | 20 |
| Tropical Atlantic box 2 (TAB2) | −70 | 10 | −5 | 15 |

# 3 Evaluation of the performance of the GCMs in simulating the rainfall in Guinea Coast and SST in ATL3 region

## 3.1 Guinean coast rainfall: annual cycle, variability and JAS mean

This section is focused on the performance of CMIP6 models in representing the rainfall annual cycle and variability over the Guinea Coast (the GCB region) for the period 1985-2014. Much of the CMIP6 models overestimate the GCB rainfall magnitude throughout the year. The monthly biases of the rainfall averaged over the GCB area in the 31 GCMs lie between $-3.4$ and $5.5\,\mathrm{mm}\cdot\mathrm{day}^{-1}$, relative to ERA5. The observed rainfalls show a bimodal structure, with two maxima in June and September respectively (Fig. 1 (a)). The first (second) peak in the observations ranges from 7.3 (7.2) to 7.8 (8.4) $\mathrm{mm}\cdot\mathrm{day}^{-1}$. Unlike the observations, the CMIP6 ensemble mean (EnsMean) depicts a plateau of $9\,\mathrm{mm}\cdot\mathrm{day}^{-1}$ in July and August, when much of the wet biases occur. This wet bias is present in 77 % and 61 % of the 31 CMIP6 models for July and August, respectively. In the other months, the CMIP6 EnsMean rainfall intensity is lower than in the reference (ERA5). The individual models hardly simulate the bimodal structure of the Guinean Coast rainfall. GFDL-ESM4, MCM-UA-1-0 and MIROC6 for instance depict a bimodal structure with an incorrect timing and intensity of the two peaks.

The RMSE of the models averaged over the entire annual cycle ranges between 0.7 and $2.6\,\mathrm{mm}\cdot\mathrm{day}^{-1}$. In addition, the spread of the modelled GCB rainfall annual cycle increases from April to October, when the interquartile range is between 1.5 and $3\,\mathrm{mm}\cdot\mathrm{day}^{-1}$ (Fig. A1 (a)). This spread is maximal in August and September. From November to March, the IQR of the models is lower or equal to $1\,\mathrm{mm}\cdot\mathrm{day}^{-1}$.

The CMIP5 models are also known to overestimate the rainfall in coastal areas of West Africa during the "little dry season" (Sow et al., 2020). Wainwright et al. (2019) have demonstrated that the misrepresentation of the July-August Guinean Coast rainfall in these models comes from the positive SST biases in the Atlantic Ocean, which strengthen the rising motions over the Guinea Coast and increase the rainfall. Consistently, the CMIP6 ensemble mean exhibits a positive (negative) mean SST bias in the eastern (western) part of the tropical Atlantic, as depicted in Fig. 2. This figure also shows low-level anomalous northerlies that reinforce the convergence over the Guinea Coast. This result is also consistent with the conclusions obtained from Richter and Tokinaga (2020), who analyzed the CMIP6 pre-industrial control simulations. Beyond the biases in the magnitude of the





rainfall simulated by the CMIP6 models, the phase of the annual cycle is relatively well simulated. This is shown in Fig. 1 (c),

where the correlation of the rainfall annual cycle in each CMIP6 model with that in ERA5 is above 0.9.

**Figure 1.** Annual cycle of the (a) rainfall intensity and (b) rainfall standard deviation in Guinea Coast. In (a) and (b), thick lines represent the annual cycle of the multimodel ensemble mean (dashed line), ERA5 (black line marked with a cross), CMAP (black line marked with a point), GPCP (black line marked with a triangle), and CRUTS403 (red line marked with a point). The other lines represent the annual cycle in each of the 31 GCMs. Taylor diagram of the (c) rainfall annual cycle, and (d) rainfall standard deviation annual cycle, where ERA5 is chosen as the reference. The annual cycle is computed for the 1985-2014 period.





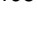

Regarding the monthly rainfall standard deviation (STD), CMAP, GPCP and CRUTS403 display a similar annual cycle, which peaks in August (Fig. 1 (b)). This maximum amounts to $1.4\,\mathrm{mm}\cdot\mathrm{day}^{-1}$ for CMAP and GPCP, and to $1.2\,\mathrm{mm}\cdot\mathrm{day}^{-1}$

for CRUTS403, while in ERA5, it is slightly lower ($1\,\mathrm{mm}\cdot\mathrm{day}^{-1}$). The annual cycle of the rainfall standard deviation over the Guinea Coast is larger in the three observed rainfall data than in ERA5 (Fig. 1 (d)). The lowest values of the monthly rainfall STD in CMAP, GPCP, CRUTS403 and ERA5 lie between 0.2 and $0.5\,\mathrm{mm}\cdot\mathrm{day}^{-1}$. The mean of the variability simulated by the CMIP6 models is higher (lower) than the variability in ERA5 during April to October (November to March). The spread of the simulated GCB rainfall STD dramatically increases from May to September, and the corresponding IQR lies between

0.6 and $0.8\,\mathrm{mm}\cdot\mathrm{day}^{-1}$ (Fig. A1 (b)). In the other months, the IQR of the GCB rainfall STD is lower than $0.3\,\mathrm{mm}\cdot\mathrm{day}^{-1}$. During July, August and September, 25, 21 and 27 GCMs overestimate the GCB rainfall STD, respectively, relative to ERA5 (Fig. 1 (b)). The correlations of the month-to-month changes of the GCB rainfall STD between models and ERA5 range from 0.5 to 0.9. The biases of the rainfall STD in the different months for the various GCMs are between $-0.6$ and $2.0\,\mathrm{mm}\cdot\mathrm{day}^{-1}$, relative to ERA5.


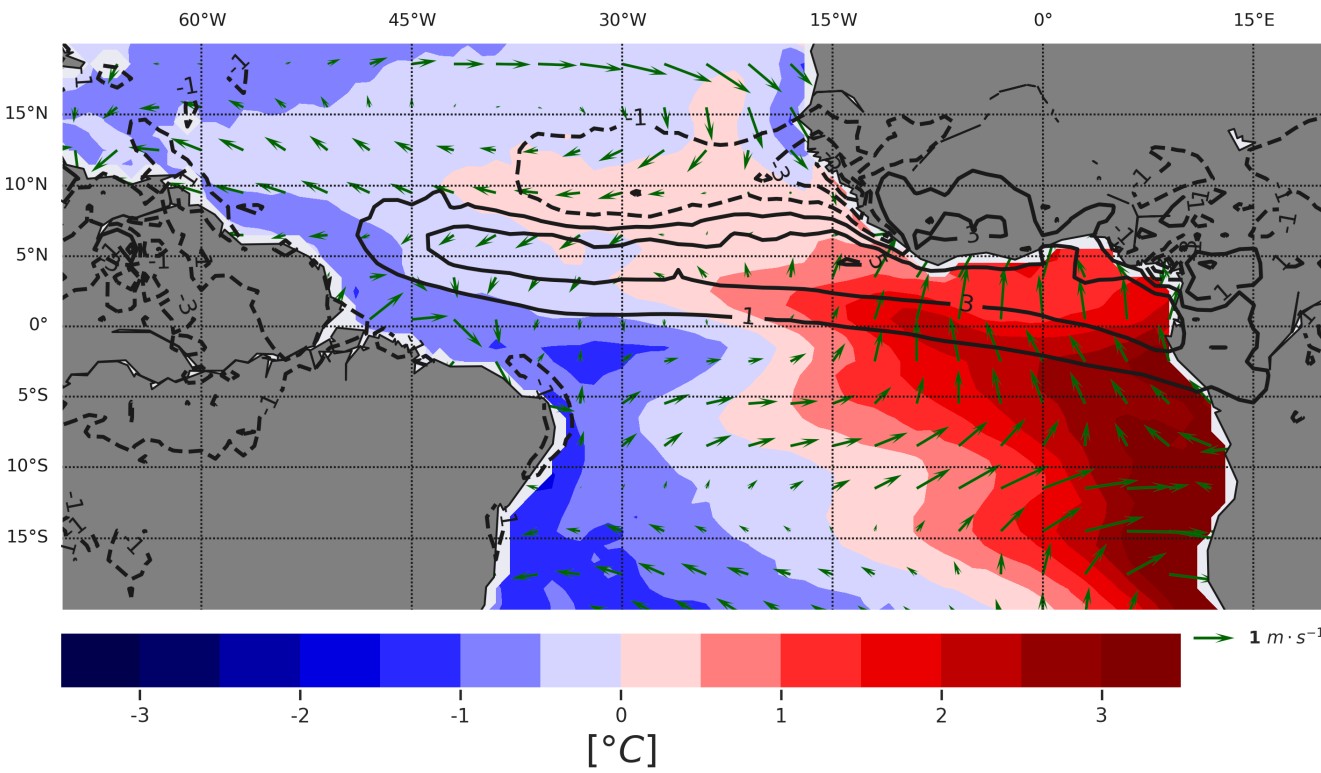

**Figure 2.** Mean biases (relative to ERA5) of the ensemble mean of the 31 GCMs for the JAS SST (in colors), rainfall (in contours) and 850 hPa wind (arrows) over 1985-2014.





In general, the JAS spatial structure of the West African rainfall is well reproduced by the models. The spatial correlation values with ERA5 lies between 0.7 and 0.95 (Fig. A2). However, the spatial variability of the rainfall intensity is underestimated in some of the models (e.g., GISS-E2-1-G and CAMS-CSM1-0), while it is exaggerated in others (e.g., GFDL-ESM4 and MIROC6).

## 3.2 SST in the ATL3 region: annual cycle, variability and JAS mean

The seasonal evolution of the SST in the Atlantic Niño region is similar in ERA5, HADISST, ERSST and COBE (Fig. 3 (a)). The annual cycle of the SST in this area shows a cold tongue which develops from April when the mean SST over the ATL3 region is about $29\,°C$ and reaches its lowest value in August, around $24.5\,°C$. The CMIP6 EnsMean overestimates the SST in this region, with a bias that ranges from $0.7\,°C$ (in October) to around $2\,°C$ (in July). This warm bias in the seasonal cycle is present in more than 87 % of the 31 CMIP6 models whatever the season, and, is more pronounced in June-September. The spread of the simulated ATL3 SST in the 31 GCMs (Fig. A3 (a)) is particularly low (high) in December (September), with an IQR equals to $0.5\,°C$ ($0.9\,°C$). The correlation between the annual cycle values in each model with ERA5 ranges from 0.88 to 0.99 (Fig. 3 (c)). This result indicates that the phasing of the SST in the ATL3 region is relatively well simulated by the models.

During the year, the observed variability of SST in the ATL3 area intensifies between April and September, with the maximum of SST standard deviation occurring in June ( $0.7\,°C$ for ERA5 and $0.6\,°C$ for HADISST, ERSST and COBE, Fig. 3 (b)). There is a second peak in November, of $0.4\,°C$ in HADISST, ERSST and COBE, whereas it is observed one month later in ERA5. This corresponds to the winter Atlantic Niño which has greatly influenced the ENSO events and the rainfall in South America, mainly during decades in the mid-20[th] century (Hounsou-Gbo et al., 2020). The first maximum of the multimodel mean of the SST standard deviation is delayed one month later compared to observations, whereas the second peak occurs at the right time. Overall, the multimodel mean overestimates the SST STD in May-July and November-January, and this difference is at its maximum in June.

Throughout the year, the monthly SST standard deviation in the models covers a wide range of values, with different phasings (Fig. 3 (d)). The spread of the SST STD annual cycle simulated by the 31 GCMs is particularly large in June, July and August, and the corresponding IQR is 0.2, 0.3 and 0.2 °C, respectively (Fig. A3 (b)). In the other months, this spread is around $0.1\,°C$.

## 4 Evaluation of the performance of the GCMs in simulating the teleconnection patterns related to the summer Atlantic Niño

### 4.1 SST pattern associated with the Atlantic Niño

The spatial SST pattern characteristic of the summer Atlantic Niño is derived by regressing the JAS SST anomalies onto the standardized JAS ATL3 index (Fig. 4 (a)). The observed SST pattern shows positive regression coefficients along the equator and off coastal areas in the southeastern Atlantic Ocean. Most of the positive SST anomalies vary between 0.1 and $0.4\,°C$.





**Figure 3.** Annual cycle of the (a) sea surface temperature and (b) sea surface temperature standard deviation in the Atlantic Niño area. In (a) and (b), thick lines represent the annual cycle of the multimodel ensemble mean (dashed line), ERA5 (black line marked with a cross), HADISST (black line marked with a point), ERSST (black line marked with a triangle), and COBE (red line marked with a point). The other lines represent the annual cycle in each of the 31 GCMs. Taylor diagram of the (c) SST annual cycle and (d) SST standard deviation annual cycle, where ERA5 is chosen as the reference. The annual cycle is computed for the 1985-2014 period.

Note that the ocean surface warming associated with positive ATL3 phases is stronger in ERA5 than in HADISST, ERSST and COBE data.



The CMIP6 models exhibit various SST imprints related to the Atlantic Niño which can be divided into two groups, according to the sign of the pattern over the tropical Atlantic (the TAB1 region, see Table 2). In the first group, 14 models display a uniform sign of the SST regression coefficients over TAB1. In the other group, in addition to the positive regression coefficients in the eastern equatorial Atlantic, the models feature negative regression coefficients in the North Atlantic (e.g., ACCESS-ESM1-5), Southwest Atlantic (e.g., HadGEM3-GC31-LL), or both North and Southwest Atlantic (e.g., MIROC-ES2L). Except for GISS-E2-1-G, the spatial correlation between the ATL3 SST pattern in models and ERA5 over the TAB1 region ranges from 0.5 to 0.9 (Fig. 4 (b)). The standard deviation of the spatial SST regression coefficients related to the ATL3 in the TAB1 region amounts $0.10\,^\circ$C in ERA5, $0.09\,^\circ$C in HADISST and $0.08\,^\circ$C in ERSST and COBE. In 29 models, this spatial variability of the anomalous SST pattern is 0.01 to $0.19\,^\circ$C higher than that in ERA5. In contrast, the spatial variability of MCM-UA-1-0 and GISS-E2-1-G SST regression coefficients is $0.02\,^\circ$C lower than that in ERA5.

## 4.2 Rainfall pattern associated with the Atlantic Niño

Figure 5 (a) displays the regression maps of the rainfall anomalies onto the standardized ATL3 index, computed for the 1985-2014 period. In the observations, the Atlantic Niño positive phases limit the northward progression of the West African monsoon flow, which leads to an anomalous increase of the rainfall over the Guinea Coast. The positive rainfall pattern associated with the Atlantic Niño extends to the ocean, from the equator up to $10\,^\circ$N and from the West African coast to $45\,^\circ$W. The spatial structure of the rainfall regression pattern is similar in CMAP-HADISST (rainfall anomalies are from CMAP, and the ATL3 index is from HADISST) and GPCP-ERSST, which are a little different from the rainfall pattern in ERA5 (Fig. 5 (a)-(b)). Moreover, while the rainfall regression coefficients in the observations and ERA5 are generally positive over the ocean in the TAB2 area (see Table 2), 23 models depict a dipolar structure of the rainfall pattern, consistent with Mohino and Losada (2015). In these models, negative values of the rainfall regression coefficients are present north of the band of positive values, between 5 and $15\,^\circ$N.

The sign-dependent average of the 1985-2014 rainfall regression coefficients over the Guinea Coast box is positive in the observations and ERA5 (Fig. A5 (a)). Two groups of GCMs are considered, according to their sign-dependent average over the GCB area. The first group is termed GC+, and defines models with a positive sign-dependent average. It includes 24 models which are able to reproduce an increased rainfall associated with the Atlantic Niño positive phases (Fig. 6 (a)). In contrast, there are 6 models in the second group, termed GC-, which show negative correlations between the rainfall over the Guinea Coast and the Atlantic Niño (Fig. 6 (b)). The GC- ensemble mean rainfall response over the GCB is weak in magnitude, with less model agreement compared to the GC+ ensemble mean. In both groups, the dipolar structure of the rainfall pattern is similar over the tropical Atlantic (the TAB2 area). However, this dipolar structure is not present in ERA5 (Fig. 6 (d)), nor in the observations.

The SST and sea surface height imprints related to the Atlantic Niño mode are similar in the GC+ and GC- ensemble means (Fig. 6 (e)-(f), (u)-(v)). In the eastern equatorial Atlantic, the core of the anomalous warming is relatively well represented in both GC+ and GC-, compared to ERA5 (Fig. 6 (h)). Contrastingly, the positive SSH anomalies in GC+ and GC- are stronger



**Figure 4.** (a) Regression maps of the JAS SST anomalies onto the standardized JAS ATL3 index. Stippling in the EnsMean indicates grid points where more than 50 % of the models show significant regression at 95 % confidence level and more than 80 % of the models agree on the sign of the regression coefficient. Stippling in each model, observations and ERA5 indicates significant regression coefficients at 95 % level. (b) Taylor diagram of the SST pattern in (a) over the tropical Atlantic box 1 where ERA5 is used as the reference.

than in ORAS5 (Fig. 6 (x)). In the western boundaries of the basin and along the $5°$N - $10°$N band, the negative SSH anomalies

265 are higher in GC+ and GC- than in ORAS5. In association with the ATL3, there is a stronger zonal moisture flux from the equatorial Atlantic toward the Guinea Coast in GC+ than in GC-. This could partially explain the rainfall response difference between the two groups (Fig. 6 (i)-(j)). The northward extension of the positive zonal moisture flux is limited to $10°$N in GC+ and $5°$N in GC-, while it reaches $15°$N in ERA5 (Fig. 6 (l)). The positive anomalous zonal moisture flux is limited to the oceanic area in GC-, whereas it extends to the Guinea Coast in GC+ and covers both the Guinea Coast and part of the Sahelian





**Figure 5.** (a) Regression maps of the JAS rainfall anomalies onto the standardized JAS ATL3 index over 1985-2014. Stippling in the EnsMean indicates grid points where more than 50 % of the models show significant coefficients at 95 % level and more than 80 % of the models agree on the sign of the regression coefficient. Stippling in each model, observations and ERA5 indicates significant regression coefficients at 95 % level. (b) Taylor diagram of the spatial JAS rainfall pattern obtained in (a) over the tropical Atlantic box 2 (black box in the maps in (a)). (c) Same as in (b) but for the West African box (green box in the maps in (a)). ERA5 is used as the reference in (b) and (c). Note that in (b), results for the couple CRUTS403-COBE is not available, as rainfall covers only lands in CRUTS403 data.

region in ERA5. Off the Guinea Coast, the meridional component of the moisture flux is also slightly greater in GC+ than in GC-, but the difference is lower than that in the zonal component (Fig. 6 (m)-(n)). However, the positive meridional moisture





flux over the equatorial and South Atlantic is weaker in both GC+ and GC-, compared to ERA5 (Fig. 6 (p)).

Finally, during positive phases of the Atlantic Niño, over the Guinea Coast and the equatorial Atlantic, the atmospheric convection is more enhanced in the GC+ group than in GC-. This is indicated by the more pronounced wind divergence difference
between the $200\,\mathrm{hPa}$ and $850\,\mathrm{hPa}$ levels in Fig. 6 (q)-(r). Moreover, in GC+ and GC-, the oceanic areas where the atmosphere is destabilized are located $5°$ south of their positions in ERA5 (Fig. 6 (t)). This explains why the positive rainfall anomalies associated with the positive phases of the Atlantic Niño in the GCMs are located south of the obtained position in ERA5. In conclusion, the combination of a large moisture flux from the equatorial Atlantic toward the Guinea Coast, and a more destabilized atmosphere over the Guinea Coast leads to an enhanced rainfall response to ATL3 in the GC+ models compared to GC- models.


Over the West African region, the latitudinal position of the maximum of the rainfall regression coefficients averaged between $20°$W and $10°$E varies in the different observational data and ERA5. This position is $4.5°$N in CMAP-HADISST, $5.5°$N in CRUTS403-COBE and $6.5°$N in GPCP-ERSST and ERA5. 75 % of the 31 GCMs depicts a maximum rainfall regression coefficient which is positioned between the West African coast and $7°$N, leading to a relatively good CMIP6 EnsMean
position at $4.5°$N (Fig. A4)). Furthermore, the spatial distribution of the rainfall regressed coefficients onto the ATL3 index in the models and ERA5 are compared. Over the West Africa (the WAB, see Table 2), the performance of the GCMs is poor to modest, which is in line with the various rainfall responses described above. The spatial correlation with ERA5 lies between $-0.6$ and $0.6$. There are 18 GCMs which present a greater spatial variability of the rainfall pattern than the one observed in ERA5 (Fig. 5 (c)). In particular, the spatial variability in 10 GCMs is one time and half greater than that in ERA5.
Over the oceanic area (the TAB2), the position of the maximum of the rainfall regression coefficients averaged between $70°$W and $10°$E ranges from 2.5 to $4.5°$N in 50 % of the 31 models (Fig. A4). The position in the CMIP6 EnsMean is located at $3.5°$N, which is two (three) degrees below the position in ERA5 (CMAP-HADISST and GPCP-ERSST). Relative to ERA5, the standard deviation of the rainfall regression coefficients onto ATL3 in the TAB2 region is generally too high in the CMIP6 models (27 models out of 31), as indicated in Fig. 5 (b). This spatial variability of the rainfall pattern in 17 GCMs is two times
greater than that in ERA5. The models show a poor to modest spatial correlation with ERA5, which ranges from $-0.4$ to $0.6$. The 1985-2014 sign-dependent average of the rainfall pattern associated with ATL3 over the equatorial Atlantic (the EAB region, see Table 2) indicates that 30 GCMs out of 31 depict an overall positive rainfall correlation with the Atlantic Niño index (Fig. A5 (b)), with only GISS-E2-1-G presenting insignificant correlations. We define a third group, OC+, which comprises these models. The aspects of the OC+ patterns are obviously very similar to the ones of the GC+ (Fig. 6 (c), (g), (k), (o), (s),
(w)).

Besides, we found no clear relation between the intensity of the warm SST biases and the JAS mean rainfall biases over the equatorial Atlantic region (the EAB) in the different models. It is also the case for the link between the warm biases and the biases of the rainfall regression coefficients related to the Atlantic Niño (Fig. A7). This conclusion is valid for the relationship
between the SST biases in the EAB region and the Guinean Coast seasonal rainfall and rainfall associated with ATL3.

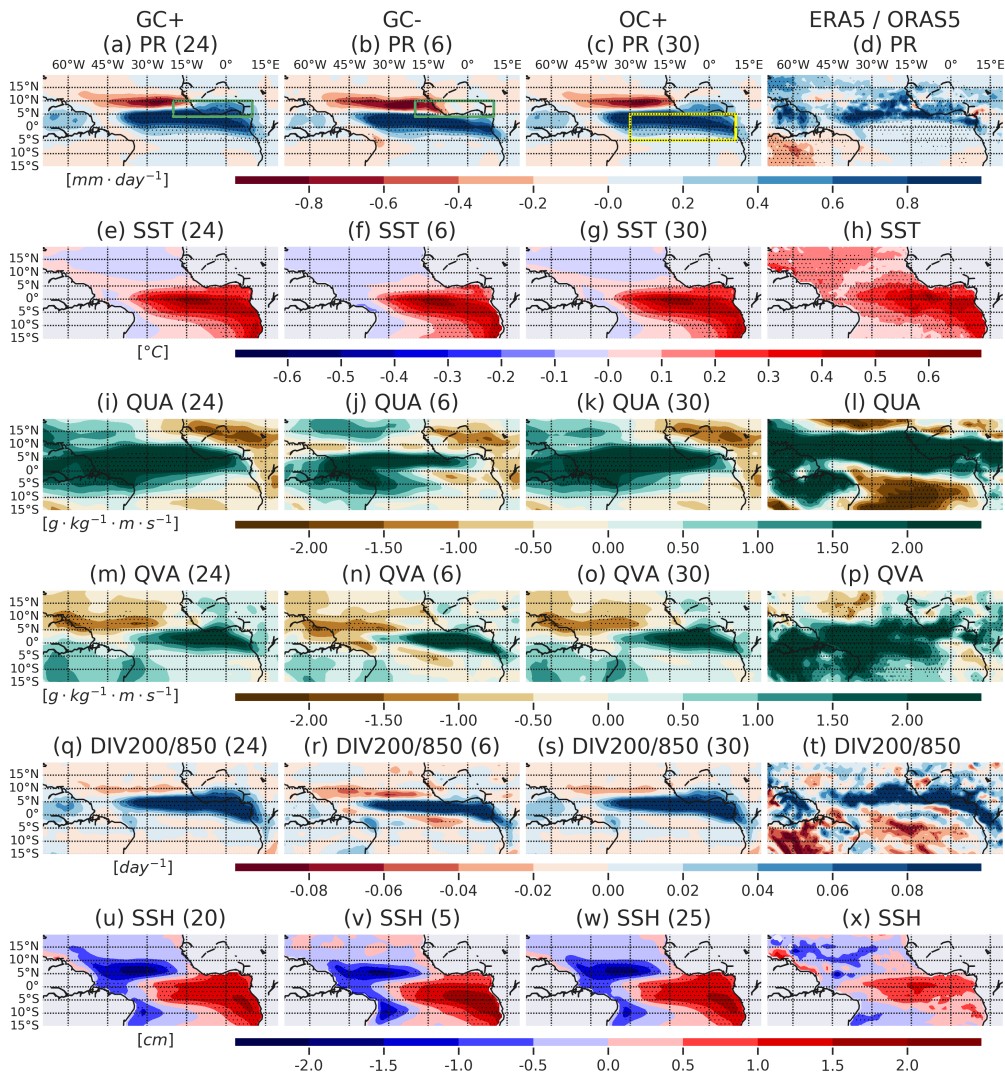

**Figure 6.** Regression maps of the JAS rainfall (a,b,c,d), SST (e,f,g,h), zonal moisture flux (i,j,k,l), meridional moisture flux (m,n,o,p), horizontal wind divergence difference between 200 hPa and 850 hPa (q,r,s,t) and sea surface height (u,v,w,x) anomalies associated with the standardized ATL3 index over the 1985-2014 period. Results for the GC+, GC- and OC+ categories are displayed, and the number of models in each category is indicated in parentheses. GC+ (GC-) and OC+ are groups of models with a positive (negative) sign-dependent average of the rainfall associated with the Atlantic Niño over the Guinea Coast (green boxes in (a) and (b)) and the equatorial Atlantic (yellow box in (c)). Results for the reanalyses are displayed in the righternmost column, and stippling represents significant regressions at 95 % level. Stippling in the other maps indicates areas where the regression coefficients are significant at 95 % confidence level for at least 50 % of the models in each group, and where more than 80 % of the models agree on the sign of the regression coefficient.





# 5  Future impact of the Atlantic Niño on the rainfall over the tropical Atlantic and Guinea Coast

## 5.1  Overview of the near-term, mid-term and long-term Atlantic Niño changes

In this section, the impact of the Atlantic Niño on the tropical Atlantic and Guinea Coast is evaluated in a climate with a high anthropogenic emission of greenhouse gases. The standard deviation of the Atlantic Niño index in the present-day climate
simulations (observations) varies between 0.24 (0.34) and 0.65 (0.38) °C (Fig. 7 (a)). Relative to the 1985-2014 period, the standard deviation of the Atlantic Niño index shows a change that ranges from -44 % to 49 % for the near-term period, -41 % to 21 % for the mid-term period and -49 % to 10 % for the long-term period (Fig. 7 (b)). The average of the relative changes in the 31 GCMs amounts to -8 %, -13 % and -21 % for the near-term, mid-term and long-term periods, respectively. Among the 31 GCMs considered, 20, 22, and 26 agree on the reduction of the Atlantic Niño variability for the near-term, mid-term
and long-term periods, respectively. Interestingly, these results are opposite to the findings of Brierley and Wainer (2018), who evaluated the change of the ATL3 variability in a quadrupled $CO_2$ experiment with CMIP5 models. They found that 3 (12) GCMs out of 15 depict a decrease (an increase) of the Atlantic Niño variability.

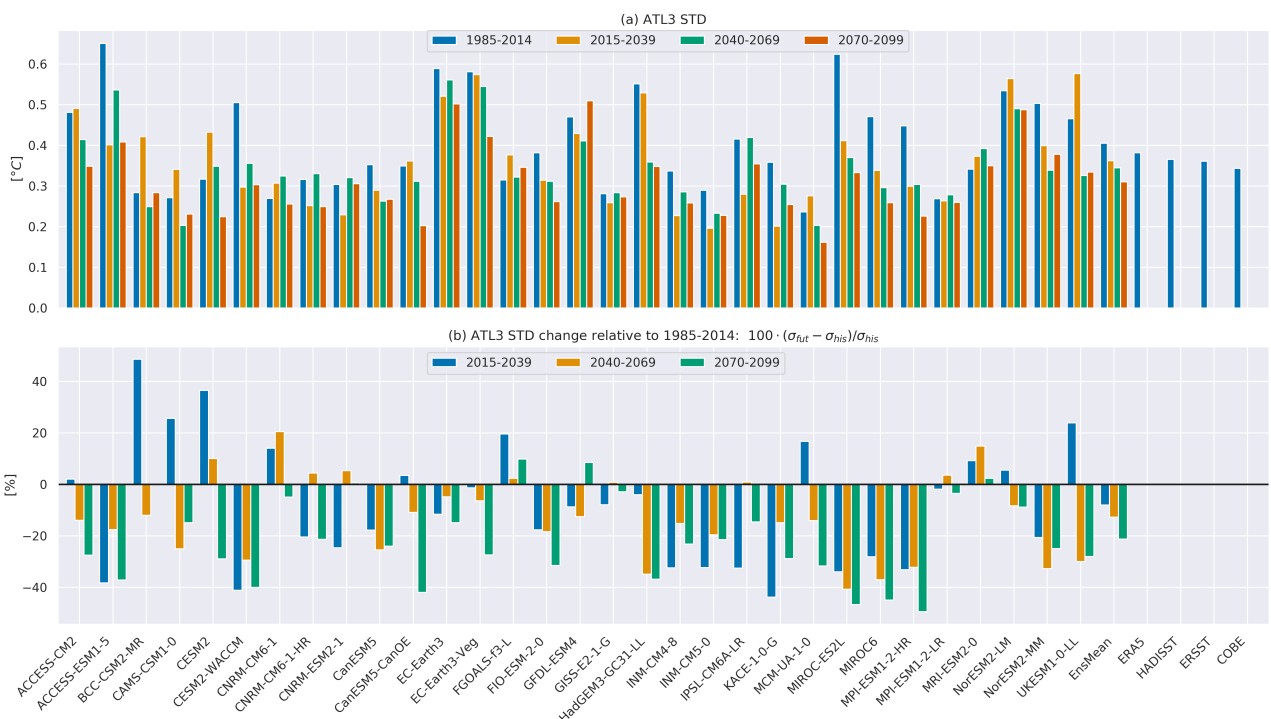

**Figure 7.** (a) Standard deviation of the JAS Atlantic Niño index over the 1985-2014, 2015-2039, 2040-2069 and 2070-2099 periods for the 31 GCMs. Results for observations and ERA5 are also shown for the 1985-2014 period. (b) Near-term (2015-2039), mid-term (2040-2069) and long-term (2070-2099) changes of the JAS Atlantic Niño standard deviation relative to the present-day period (1985-2014) for each of the 31 GCMs.



In the present-day climate, the multimodel mean of SST anomalies associated with ATL3 and averaged over the equatorial

Atlantic (the EAB region) is $0.32\,^{\circ}$C. This value is consistent with the observations and ERA5. For the periods 2015-2039,
2040-2069 and 2070-2099, the EnsMean value is reduced to 0.29, 0.28 and $0.26\,^{\circ}$C, respectively. This result indicates that,
relative to the present-day climate, the multimodel mean of the SST response to the Atlantic Niño has gradually decreased, with
a percentage of change equals to -8 %, -13 % and -19 % for the near-term, mid-term and long-term periods, respectively (Fig. 8
(a)). There are 81 % and 84 % of the 31 GCMs which agree on the sign of the change in the mid and long-term periods, against

55 % in the near-term period. In addition, the consistency of the equatorial Atlantic SST response to ATL3 among the GCMs
gradually increases with time, as highlighted by the shrinking of the interquartile range of the models SST anomalies, from a
value $0.13\,^{\circ}$C in 1985-2014, to $0.07\,^{\circ}$C in 2070-2099. These results are in line with the changes of the standard deviation of
the ATL3 index.

Over the period 1985-2014, the CMIP6 EnsMean rainfall anomalies related to one standard deviation of the ATL3 index and

averaged over the EAB is $0.59\,\mathrm{mm\cdot day^{-1}}$, which is greater than the observed value ($0.2\,\mathrm{mm\cdot day^{-1}}$). Subsequent to the
weakening of the equatorial SST anomalies in the future, the rainfall associated with ATL3 over the EAB area decreases (Fig.
8 (b)). The CMIP6 EnsMean of the anomalous EAB rainfall values for the three consecutive future periods are 0.50, 0.44 and
$0.37\,\mathrm{mm\cdot day^{-1}}$, respectively. The corresponding EnsMean rainfall reduction relative to the present-day period is about 14 %,
25 % and 37 %, with an agreement of 68 %, 77 % and 84 % of the models on the sign of the change in the near-term, mid-term

and long-term periods, respectively.

Figures 9 (a) and (c) indicate that, over the Atlantic Ocean, in the four periods of the study, the position of the maximum
rainfall anomalies associated with ATL3 remains close to $4\,^{\circ}$N, but with a linear decrease of the rainfall intensity with time.
North of $5\,^{\circ}$N, an upper level subsidence leads to negative rainfall anomalies in the tropical North Atlantic, which gradually

weaken in the future periods. Over the EAB region, the decrease of the rainfall interquartile range in the CMIP6 models for the
mid and long-term periods indicates an increased consistency of the rainfall response among the models. For the present-day,
the near-term, the mid-term and the long-term periods, the IQR values are 0.46, 0.49, 0.40 and $0.34\,\mathrm{mm\cdot day^{-1}}$, respectively.

Regarding the ATL3 induced rainfall over the Guinea Coast, an overall reduction of the multimodel mean is projected (Fig.

8 (c), Figs. 9 (b) and (d)). The CMIP6 EnsMean rainfall averaged over the Guinea Coast (the GCB area) decreases from a
value of $0.36\,\mathrm{mm\cdot day^{-1}}$ in the period 1985-2014 to $0.22\,\mathrm{mm\cdot day^{-1}}$ at the end of the 21st century. The corresponding values
for the near-term and mid-term periods are 0.29 and $0.24\,\mathrm{mm\cdot day^{-1}}$, respectively. The amount of the GCB EnsMean rainfall
reduction reaches thus 18 %, 33 % and 38 % in the near-term, mid-term and long-term periods, respectively. The percentages
of the models that agree on a reduction of the rainfall magnitude associated with ATL3 over GCB for the three periods are 61

%, 65 % and 58 %, respectively. This means that there is a lesser agreement on the projected ATL3-rainfall changes over the
Guinea Coast than over the equatorial Atlantic. Moreover, the IQR of the rainfall associated with ATL3 over the Guinea Coast
is more or less the same over the four periods ($0.6\,\mathrm{mm\cdot day^{-1}}$).

In the future climate, following the SSP5-8.5 scenario, the variability of the trade winds associated with the Atlantic Niño



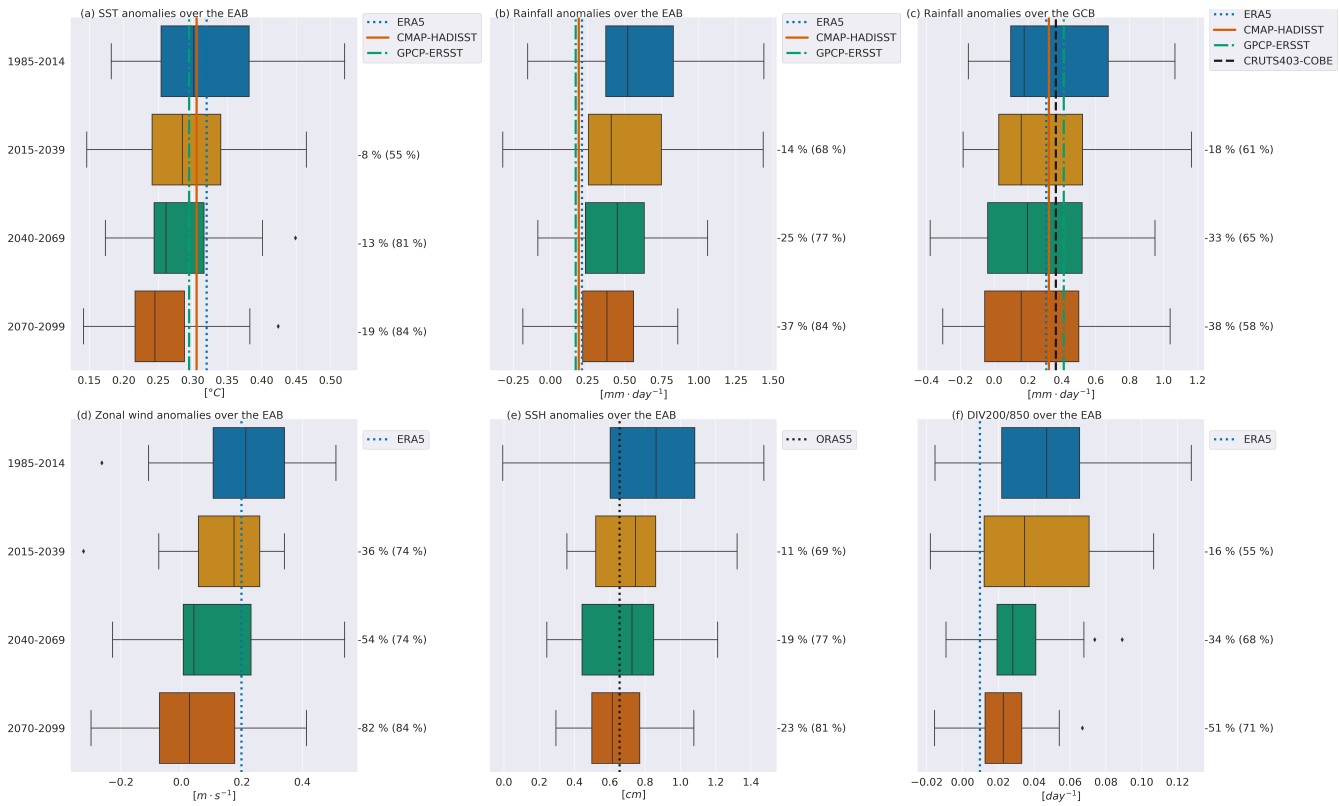

**Figure 8.** Boxplots of the regression coefficients of (a) SST, (b) rainfall, (d) 850 hPa zonal wind, (e) sea surface height and (f) horizontal wind divergence difference between 200 hPa and 850 hPa (DIV200/850) anomalies averaged over the equatorial Atlantic box. Boxplot of the regression coefficients of the rainfall anomalies averaged over the Guinea Coast (c). The variables from each of the 31 models are regressed onto the standardized JAS ATL3 index, for four different periods: 1985-2014, 2015-2039, 2040-2069 and 2070-2099. Each boxplot represents the distribution of the 31 regression coefficients averaged over the specified area in the different models. Vertical lines represent the mean values in the observations and ERA5 for the 1985-2014 period.

decreases, as shown in Figs. 8 (d) and 9 (e), where the 850 hPa zonal winds are regressed onto the standardized ATL3 index
and averaged over the equatorial Atlantic (the EAB region). This indicates a weakening of the first component of the Bjerkness
feedback, which links anomalous westerlies to the abnormal warming in the eastern equatorial Atlantic. The CMIP6 EnsMean
of the 850 hPa zonal wind anomalies corresponding to one standard deviation of the ATL3 index and averaged over the EAB
is equal to $0.22\,\mathrm{m\cdot s^{-1}}$ in the 1985-2014 period, which is close to the value derived from ERA5. For the near-term, mid-term
and long-term future, this value decreases to 0.14, 0.10 and $0.04\,\mathrm{m\cdot s^{-1}}$, respectively. This reduction corresponds to 36 %, 54
% and 82 % of the value obtained in the present-day climate, respectively, with a high agreement among the 31 CMIP6 models
(74 % for the near-term and mid-term periods, and 84 % for the long-term period).

The anomalous westerlies forcing on the eastern equatorial Atlantic thermocline is the second component of the Bjerkness





feedback. In response to weakened trade winds, the upwelling of seawater in the east is reduced, leading to an anomalous rising of the SSH in the east, and an SSH fall in the west (Fig. 8 (e) and Fig. 9 (f)). The weakening of the anomalous westerlies associated with one standard deviation of the ATL3 index in the future reduces the positive (negative) SSH anomalies in the eastern (western) equatorial Atlantic. The CMIP6 EnsMean average of the SSH anomalies related to the standardized ATL3 index over the EAB region for the four consecutive periods is 0.83, 0.74, 0.67 and $0.64\,\mathrm{cm}$, respectively. The corresponding reduction of the SSH EnsMean in the EAB for the three consecutive future periods is 11 %, 19 % and 23 %. The CMIP6 models agreement on the sign of this change is higher than 68 % in the three cases. In addition, the spread of the models is also reduced at the end of the $21^{\mathrm{st}}$ century, highlighting the increased consistency of the models for the SSH responses to ATL3.

Part of the weakening of the tropical Atlantic rainfall teleconnection associated with the Atlantic Niño in a warmer climate has been attributed to a faster warming of the mid-tropical Atlantic troposphere compared to the surface, as pointed out by Jia et al. (2019). This implies that the troposphere above the equatorial Atlantic becomes more stable and limits the convection. This is in agreement with Figure 8 (f) that shows, over the equatorial Atlantic, a gradual decrease of the strength of the rising motion associated with the Atlantic Niño warm phases, during the near-term, mid-term and long-term periods. For the three successive future periods, the relative change (and the agreement between models) of the CMIP6 EnsMean DIV200/850 averaged over the EAB area is -16 % (55 %), -34 % (68 %) and -51 % (71 %), respectively. The consistency of the DIV200/850 anomalous response in the CMIP6 models is significantly enhanced in the mid-term and long-term periods.

To highlight the impact of some specific differences between the models in their way of representing the spatial characteristics related to ATL3 in response to climate change, six different subsets of the CMIP6 models are considered. They are also based on the sign-dependent average of the ATL3 related rainfall teleconnection pattern over, on the one hand, the Guinea Coast (the GCB area) and, on the other hand, the equatorial Atlantic region (the EAB area). First, the GC+ group (the 24 models in Sect. 4.2 which simulate a realistic GCB rainfall associated with one standard deviation of the ATL3) is split into two groups. One group, termed GC+-, projects a decrease of the GCB rainfall intensity associated with the Atlantic Niño. The other group, termed GC++, project an increase of the GCB rainfall magnitude associated with the Atlantic Niño (see Fig. A5 for the values of the sign-dependent average). These two different signals are considered because we do not know exactly the realistic ATL3-rainfall patterns in the future. The six models that simulate weak or negative ATL3-rainfall anomalies over the GCB during the 1985-2014 period are similarly divided into two groups, GC- - and GC-+, which contain models projecting a strengthened and a decreased (or inverted) rainfall intensity related to one ATL3 standard deviation, respectively.

Next, nearly all of the 31 CMIP6 models (except GISS-E2-1-G) are able to simulate a positive rainfall anomaly over the Atlantic Ocean in the area between the equator and $5°$ N. The same sign-dependent average analysis performed over the equatorial Atlantic region (the EAB area) leads to the OC++ and OC+- groups. Whatever the period considered in the future, the GC+- and OC+- groups contain the largest number of models. They show an overall decrease of the rainfall magnitude related to ATL3 in the future, which is associated with a weakened Bjerkness feedback. By contrast, in GC++ and OC++ groups, weak changes of the zonal zonal wind and sea surface height anomalies indicate that the Bjerkness feedback act similarly in



**Figure 9.** Rainfall anomalies associated with ATL3 and averaged over $30\,°\text{W}$ and $10\,°\text{E}$ for oceanic areas (a) and land areas (b). Gray bands in (a) and (b) represent the equatorial Atlantic and Guinea Coast regions, respectively. Zonal variation of the rainfall anomalies associated with ATl3 and averaged over the equatorial Atlantic (c) and Guinea Coast regions (d). Zonal variation of the 850 hPa wind (e) and sea surface height anomalies (f) associated with ATL3 and averaged over the equatorial Atlantic. In each panel, the solid black line represents ERA5 or ORAS5 for 1985-2014, and the other solid lines represent the ensemble mean of the 31 GCMs for the present and future periods. The dotted, dashed-dotted, or dashed lines represent the observations for the 1985-2014 period.

the present and future climates. Moreover, the spatial characteristics of the changes are similar for the near-term, mid-term and long-term periods, with an enhancement of the pattern of change with time (as summarized in Fig. 9). Therefore, we will focus
400     the next discussion on the 2070-2099 changes relative to the 1985-2014 period, as these changes are the most amplified. The detailed lists of the models in the different categories for the three future periods are available in Tables A1, A2 and A3.





## 5.2 Change of the Atlantic Niño impact on the Guinea Coast rainfall

Given our choice by construction, and focusing on the ATL3-related long-term rainfall changes over the Guinea Coast, among
405 the 24 models in the GC+ group, 15 (9) project a decrease (an increase) of the rainfall magnitude associated with ATL3 (Fig.
10 (a)-(b)). The projected ATL3-rainfall signal in the GC+- group is limited to the Atlantic Ocean, and is hardly robust over
the Guinea Coast. However, models in the GC++ group show an eastward and northward shift of the anomalous positive rain-
fall belt over the Atlantic Ocean. This results in an enhancement of the rainfall teleconnection pattern over the Guinea Coast,
mainly over the Cameroon mountains and Guinean highlands.


Regarding the SST patterns related to the Atlantic Niño, the projected changes display a decrease in SST over the equatorial
Atlantic and off the Angola-Benguela Coast in both GC+- and GC++ (Fig. 10 (f)-(g)). However, the SST patterns of change in
GC+- and GC++ are different in the north and south of the $5^\circ$ N - $5^\circ$ S band. Compared to the 1985-2014 period, there is a
poleward and westward extension of the anomalous warming associated with ATL3 in GC+-, which directly leads to decrease
zonal and meridional surface pressure gradients over the Atlantic Ocean, and a reduced surface convergence over the equatorial
Atlantic (Fig. 10 (u)). Concerning the 850 hPa zonal wind anomalies, their magnitude associated with ATL3 in the GC+- group
is projected to decrease in the future (Fig. 10 (k)), while this change is very weak and unrobust in GC++ group (Fig. 10 (l)).
As a consequence, in the GC+- group, the projected equatorial Atlantic zonal moisture flux and moisture flux convergence are
weaker, and their contribution to increase the Guinea Coast rainfall is limited (Fig. 10 (u)).


The SSH response to ATL3 in GC+- is also projected to weaken in the future, compared to the present-day situation (Fig. 10
(p)). Together with the important change of the near surface zonal wind, these results suggest a weakened Bjerkness feedback
over the equatorial Atlantic in a warmer climate. In the GC++ group, there is no substantial difference between the 1985-2014
and 2070-2099 SSH teleconnection patterns associated the Atlantic Niño (Fig. 10 (q)). This means that the intensity of the
Bjerkness feedback remains quite similar in the GC++ models during the two periods. Furthermore, compared to the present-
day situation, in GC++, the surface moisture convergence related to the Atlantic Niño is displaced eastward and northward
over the equatorial Atlantic and Guinea Coast in the future (Fig. 10 (v)). This is consistent with the projected ATL3-rainfall
increase over the Guinea Coast in GC++, in agreement with Mohino and Losada (2015). Besides, over the equatorial Atlantic,
there is an overall decrease of the convection associated with ATL3 in both GC+- and GC++ models (Figs. A8 and A9).


From the six models in the GC- group, four (the GC-+ group) project an eastward shift of the positive ATL3-rainfall anoma-
lies over the tropical Atlantic, leading to an increased rainfall intensity over the Guinea Coast (Fig. A10). In addition, the
negative ATL3-rainfall located north of the positive ATL3-rainfall band weakens in these models. This future response is, how-
ever, unrobust over the Guinea Coast. In the other two models (the GC- - group), over the tropical Atlantic, the dipolar rainfall
response to the ATL3 remains present in the future period. Moreover, the band of negative rainfall anomalies which extends





from the ocean to the westernmost region of the Guinea Coast is shifted equatorward.



**Figure 10.** Long-term changes of the JAS rainfall (a-e), SST (f-j), 850 hPa zonal wind (k-o), sea surface height (p-t), moisture flux (vectors) and moisture flux divergence (in colors) (u-y) regression patterns associated with the standardized ATL3 index, relative to the present-day climate (2070-2099 minus 1985-2014). Stippling in (a)-(t) and contours in (u)-(y) indicate areas where the mean change (in colors) is significant at 95 % level according to a two-sided Welch t-test and where at least two thirds of the models agree on the sign of the change. The number of models in each group is indicated in parentheses.

### 5.3 Change of the Atlantic Niño impact on the equatorial Atlantic rainfall

When considering the ATL3-rainfall pattern over the equatorial Atlantic, by construction, 21 GCMs belonging to the OC+-
group project a decreased rainfall signal in a warmer climate (Fig. 10 (c)). In contrast, 9 GCMs in the OC++ group show no





change of the ATL3-rainfall magnitude (Fig. 10 (d)). The change patterns of the ATL3-SST anomalies are similar in the OC+- and OC++ groups. They show a future decrease of the SST associated with ATL3 in the Atlantic Niño region (Fig. 10 (h)-(i)). However, while there is no change in the low level zonal wind response to ATL3 in the OC++ group, the OC+- group projects an important weakened zonal wind speed related to ATL3 (Fig. 10 (m)-(n)).


The analysis of the SSH anomalies associated with one standard deviation of the ATL3 reveals a projected decrease of the SSH in the tropical Atlantic for the OC+- group (Fig. 10 (r)). In the OC++ group, this change is weaker (Fig. 10 (s)). These results suggest a projected weakened Bjerkness feedback in the OC+- group, while this feedback keeps the same intensity in the OC++ group. Furthermore, consistent with the change of the rainfall pattern, Figure 10 (w) indicates a decrease of the low

level moisture flux and moisture flux convergence associated with ATL3 over the equatorial Atlantic in the OC+- group. This change is weaker in OC++ compared to OC+- (Figs. 10 (w)-(x)). In both OC+- and OC++, there is a projected decrease of the convection over the equatorial Atlantic during the ATL3 positive phases, as in GC+- and GC++ groups (Figs. A8 and A9). Overall, the patterns of change of the 31 CMIP6 EnsMean are similar to those in OC+- and GC+- groups (Figs. 10 (e), (j), (o), (t), (y)).


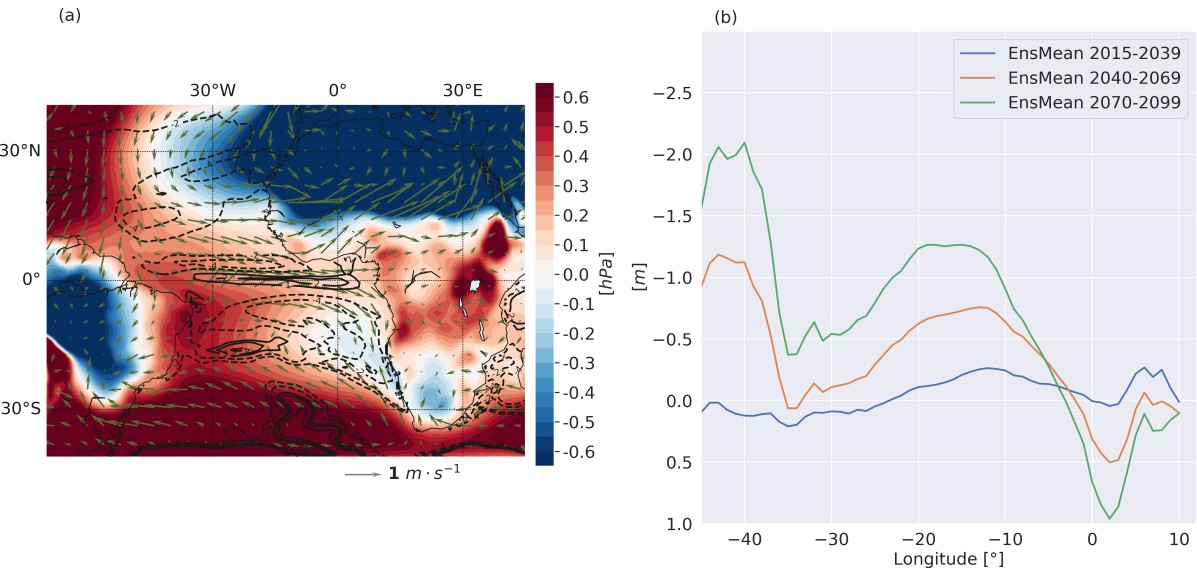

**Figure 11.** (a) Long-term mean state change (2070-2099 minus 1985-2014) of the sea level pressure (in colors), 10 m horizontal wind (arrows) and mixed layer depth (in contours in m). (b) Mean change of the mixed layer depth averaged over the equatorial Atlantic (the EAB region), for the near-term, mid-term and long-term periods, relative to the present-day conditions. These changes are computed for the JAS season and averaged over 23 GCMs (Table 1).





In a warmer climate, the deepening of the Saharan heat low and its extension over the North Atlantic weakens the mean equatorial easterlies in JAS. In addition, over the tropical Atlantic, the models project an anomalous increase (decrease) of the mean sea level pressure (SLP) in the western (eastern) part of the basin in the future. This favors anomalous westerlies over the equatorial Atlantic (Fig. 11 (a)). As a consequence, the weakened trade winds lead to a deeper mean thermocline in the

eastern equatorial Atlantic. This is indicated by the increased oceanic mixed layer depth, here used as a proxy for the top of the thermocline depth. Figure 11 (b) shows that this deepening is gradual with time. As the thermocline becomes deeper, it responds less to the SST changes at the surface during the Atlantic Niño phases, and leads to a weakened Bjerkness feedback.

## 6   Discussion and Conclusion

In this study, the seasonality of the rainfall in the Guinea Coast has been analyzed based on the output of 31 models partici-

pating in the CMIP6 project. First, the 1985-2014 period of the historical simulations has been considered. Results show that the majority of the models fails to reproduce the bimodal structure of the rainfall annual cycle over this area, due to a warm surface bias in the eastern tropical Atlantic. The amplitude of the Guinean Coast rainfall annual cycle is also larger than in the observations, owing to a wet bias in this area in boreal summer. Overall, the Guinean Coast rainfall annual cycles in the CMIP6 models are reasonably well correlated with the one in the reanalysis, ERA5. The correlation values are above 0.9. Regarding

the annual cycle of the rainfall standard deviation, models display a wide range of variability. Averaged over the entire annual cycle, this variability is more larger in the models than in ERA5. The correlations between the models and ERA5 rainfall STD annual cycles range from 0.5 to 0.9.

The annual cycle of the ATL3 index and the spatial pattern of the associated SST anomalies in the GCMs show good corre-

lations with the observations and ERA5. However, the maximum rainfall position associated with the Atlantic Niño in the 31 GCM ensemble mean is displaced south of the observed positions over the tropical Atlantic. Nearly all the models manifest an increased rainfall associated with positive ATL3 phases over the equatorial Atlantic. In the case of the Guinea Coast region, 24 models exhibit a realistic increased rainfall related to a warm phase of the Atlantic Niño, against 6 models which show a negative or weak rainfall response. In connection with ATL3, there is a weaker zonal wind response and a weaker zonal

moisture flux toward the Guinea Coast, as well as a weaker deep convection in these 6 models (the GC- group), relative to the 24 models (the GC+ group).

Secondly, the change of the Atlantic Niño and its impact on the rainfall over the equatorial Atlantic and the Guinea Coast in a climate with a high anthropogenic emission of greenhouse gases has been evaluated. Analyses of the Shared Socioeconomic

Pathway 5-8.5 simulations demonstrate a gradual decrease of the SST anomalies associated with one standard deviation of the Atlantic Niño index for the near-term, mid-term and long-term periods, relative to the present-day climate. The ensemble mean change of the standard deviation of the ATL3 index relative to the present-day period is $-8$ %, $-13$ % and $-21$ % for those three periods, respectively. Among the 31 GCMs, 20, 22 and 26 agree on the sign of the ATL3 STD change, for the three future





periods respectively. Much of the models also indicate a gradual weakening with time of the rainfall magnitude related to the
Atlantic Niño over the Guinea Coast and the equatorial Atlantic in the future.

More specifically, focusing on the Guinea Coast, among the 24 models which present a relatively good ATL3-rainfall signal
in the 1985-2014 period, there are 15 GCMs (the GC+- group) which project a relative decreased rainfall related to ATL3 in
the 2070-2099 period. Analyses of the SST, 850 hPa zonal wind and SSH anomalies indicate that the magnitude of their varia-
tions associated with one standard deviation of the Atlantic Niño decreases, suggesting a weakening of the Bjerkness feedback
in a warmer climate. By contrast, 9 models show weak differences in the patterns associated with the Atlantic Niño, and a
little increase in the rainfall pattern over elevated areas in Guinea Coast (the GC++ group). While the GC+- group exhibits an
important decrease of the zonal wind variability associated with the Atlantic Niño in the future, this variability remains similar
in the GC++ group. In these models, we argue that the Bjerkness feedback should be of the same intensity in the present-day
and at the end of the 21$^{st}$ century. Alongside, there is a projected westward and a poleward extension of the anomalous warm
water associated with the Atlantic Niño in the GC+- group. This contributes to the reduction of the equatorial Atlantic surface
convergence associated with ATL3.

Similarly, over the tropical Atlantic region, 30 models out of the 31 show a dipolar ATL3-rainfall structure, with positive
anomalies between the equator and $5\,^{\circ}$N and negative anomalies north of this band. For the end of the 21$^{st}$ century, 21 models
project a reduced rainfall magnitude associated with the Atlantic Niño. The changes in this group of models (OC+-) are sim-
ilar to the ones in the GC+- group. Likewise, 9 models exhibit no substantial change of the ATL3-rainfall over the equatorial
Atlantic. In these models, unlike the OC+- ones, the zonal wind response to ATL3 is more or less identical in the present-day
and future periods. The Bjerkness feedback is projected to weaken in the OC+- group, while its intensity remains the same in
the OC++ group.

In the GC+- and OC+- groups, the equatorial Atlantic convection associated with the Atlantic Niño is projected to decrease,
as indicated by the weakened wind divergence difference between 200 hPa and 850 hPa. This decrease is weaker in GC++ and
OC++ groups. This result adds support to Jia et al. (2019), who found a more stabilized troposphere over the tropical Atlantic
in the future, which in turn reduces the convection during the positive phases of the Atlantic Niño. Their analysis was focused
on the link between the Atlantic Niño and La-Niña, while our analysis is focused on the Atlantic Niño teleconnection with the
tropical Atlantic and Guinea Coast.

In the last part of this study, we linked the background JAS mean state change to the change of the Atlantic Niño variabil-
ity. In a warmer climate, we found an oceanic extension of the Saharan Heat Low over the North Atlantic. This anomalous
low pressure weakens the equatorial trade winds. In addition, over the equatorial Atlantic, the mean sea level pressure change
is positive in the western part of the basin, and decreases eastward. This contributes to decreasing the intensity of the mean
easterlies, thus limiting the equatorial upwelling. Subsequently, the thermocline becomes deeper along the equator, and the





deepening is more important in the eastern equatorial Atlantic. For this reason, the influence of the eastern equatorial Atlantic

thermocline on the surface during the Atlantic Niño phases is weakened in the future. This explains the projected decreased

variability of the Atlantic Niño in the SSP5-8.5 emission scenario.

Future studies are needed to assess the impact of the warm biases in the eastern equatorial Atlantic on the teleconnection

patterns of the Atlantic Niño, and could help to increase the reliability of the projected changes. Moreover, the reduction

of these biases is crucial for the rainfall projections over West Africa. Further analyses are also needed to understand the

multidecadal modulation of the Guinean Coast rainfall and extreme rainfall by oceanic internal modes of variability in climate

models for both present-day and future climate conditions.

*Author contributions.* KW, HG and TF conceptualized the paper. KW performed the analyses and prepared the figures. FK participated in
the discussions. KW, HG and TF wrote the manuscript. All the co-authors provided scientific inputs and commented on the final draft.

*Competing interests.* The authors declare that they have no conflict of interest.

*Acknowledgements.* We acknowledge the World Climate Research Programme, which, through its Working Group on Coupled Modelling,
coordinated and promoted CMIP6. We thank the climate modelling groups for producing and making available their model output, the Earth
System Grid Federation (ESGF) for archiving the data and providing access, and the multiple funding agencies who support CMIP6 and
ESGF.





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





# Appendix A

## 720  A1  Appendix figures

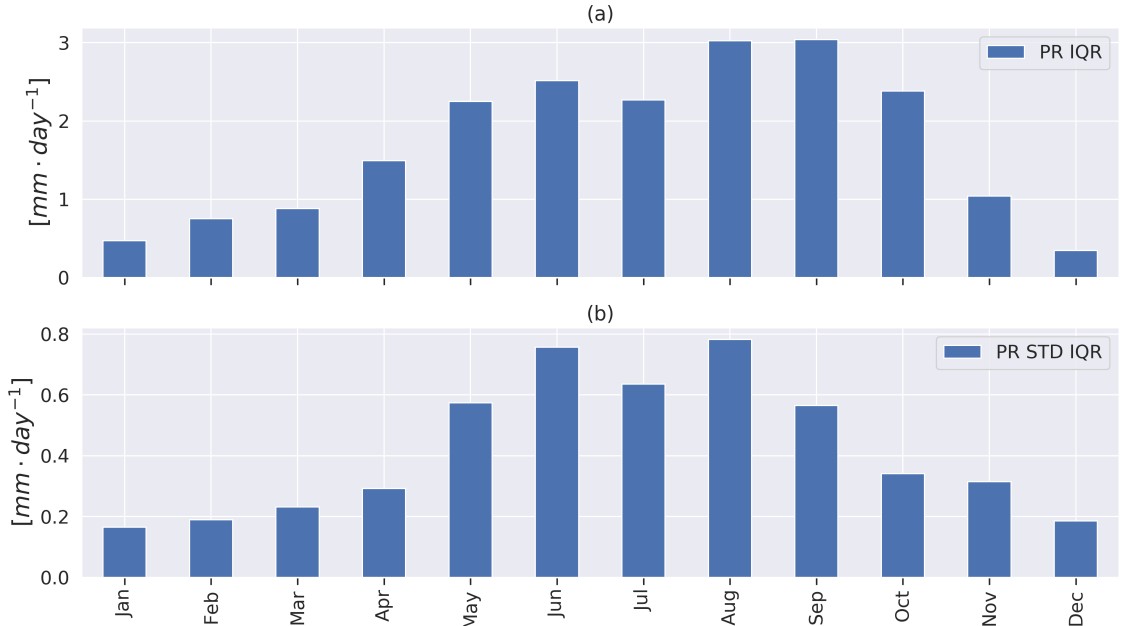

**Figure A1.** Interquartile ranges of (a) the Guinean Coast rainfall annual cycle and (b) the Guinean Coast rainfall standard deviation annual cycle simulated by 31 GCMs participating in the CMIP6 project. The analysis is based on the 1985-2014 period.



**Figure A2.** (a) 1985-2014 JAS rainfall mean in 31 GCMs, in ERA5, CMAP, GPCP and CRU. (b) Taylor diagram of the spatial JAS rainfall mean over West Africa (4°N-20°N), where the mean JAS rainfall patterns in the models and observations are compared to the one of ERA5.



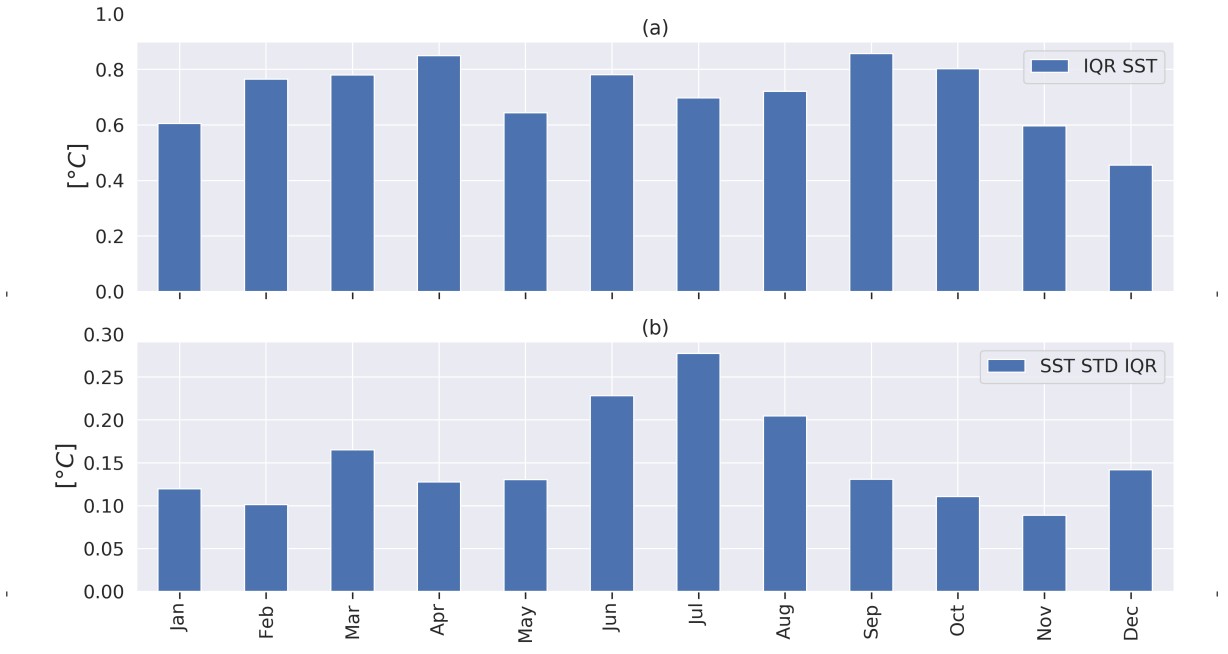

**Figure A3.** Interquartile range of the (a) ATL3 SST annual cycle and (b) ATL3 SST standard deviation annual cycle simulated by 31 GCMs participating in the CMIP6 project. The analysis is based on the 1985-2014 period.





**Figure A4.** Position in latitude of the maximum rainfall associated with ATL3 in the tropical Atlantic (zonal average over $70\,^{\circ}$W and $10\,^{\circ}$E) and West Africa (zonal average between $20\,^{\circ}$W and $10\,^{\circ}$E ). The boxplots represent the distribution of the positions computed in each of the 31 models. The black line inside each box represents the median value of the models, and the outliers are represented by the diamond-shaped symbols. The other horizontal lines represent the mean positions obtained in the multimodel ensemble mean (green line), ERA5 (red dashed line), CMAP-HADISST (blue dashed-dotted line), GPCP-ERSST (black dotted line) and CRUTS403-COBE (purple dotted line).





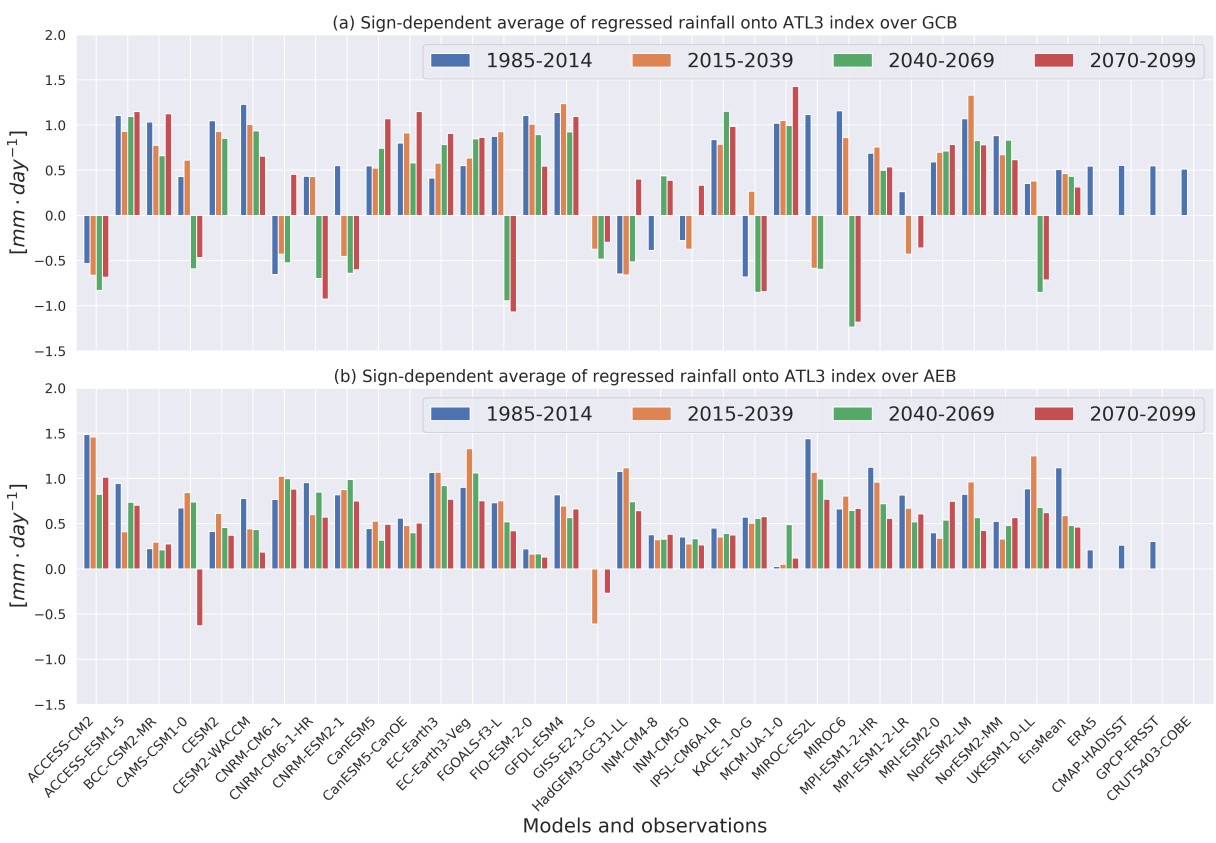

**Figure A5.** Sign-dependent average of the JAS rainfall regressed coefficients onto the standardized ATL3 index over the Guinea Coast box (a) and the equatorial Atlantic box (b) for the 31 GCMs over the present-day, near-term, mid-term and long-term periods. The sign-dependent average values in the observations and ERA5 are computed for the 1985-2014 period.



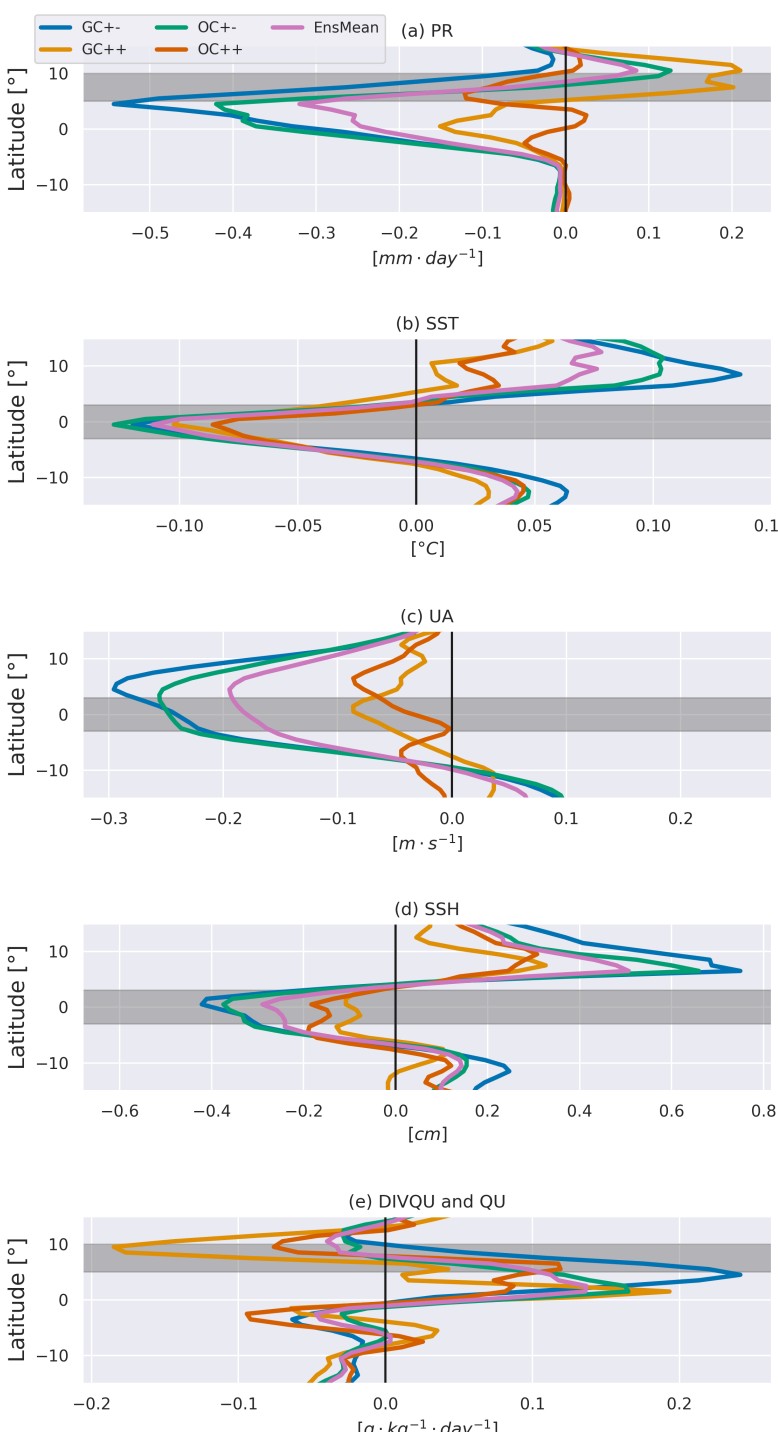

**Figure A6.** 2070-2099 minus 1985-2014 changes: Zonal mean (mean over 30°W-10°E) of (a) rainfall, (b) SST, (c) 850 hPa zonal wind, (d) SSH and (e) 850 hPa divergence of moisture flux anomalies for the GC+-,GC++, OC+-, OC++ and the 31 GCMs EnsMean groups of models. Gray bands represent the positions of the Guinea Coast (a,e) and ATL3 regions (b-d).



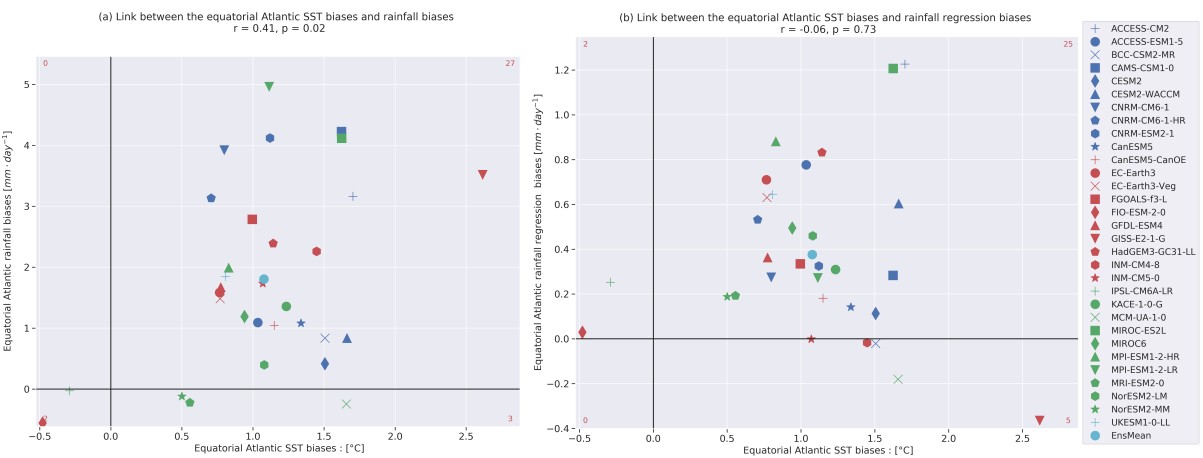

**Figure A7.** (a) Relation between the JAS mean SST biases and the JAS mean rainfall biases in the CMIP6 models. (b) Relation between the JAS mean SST biases and the biases of JAS rainfall regressions onto the ATL3 index in the CMIP6 models. Biases are computed over the equatorial Atlantic box.





**Figure A8.** 1985-2014 regression maps of rainfall, SST, zonal moisture flux, meridional moisture flux, DIV200/850, SSH, moisture flux and moisture flux divergence anomalies onto the standardized ATL3 index, for the GC+-, GC++, OC+-, OC++ and the 31 GCMs EnsMean groups. Stippling indicates areas where the regression coefficients are significant at 95 % confidence level for at least 50 % of the models in each group, and where more than 80 % of the models agree on the sign of the regression coefficient. The number of models in each group is indicated in parentheses.



**Figure A9.** 2070-2099 regression maps of rainfall, SST, zonal moisture flux, meridional moisture flux, DIV200/850, SSH, moisture flux and moisture flux divergence anomalies onto the standardized ATL3 index, for the GC+-, GC++, OC+-, OC++ and the 31 GCMs EnsMean groups. Stippling indicates areas where the regression coefficients are significant at 95 % confidence level for at least 50 % of the models in each group, and where more than 80 % of the models agree on the sign of the regression coefficient. The number of models in each group is indicated in parentheses.

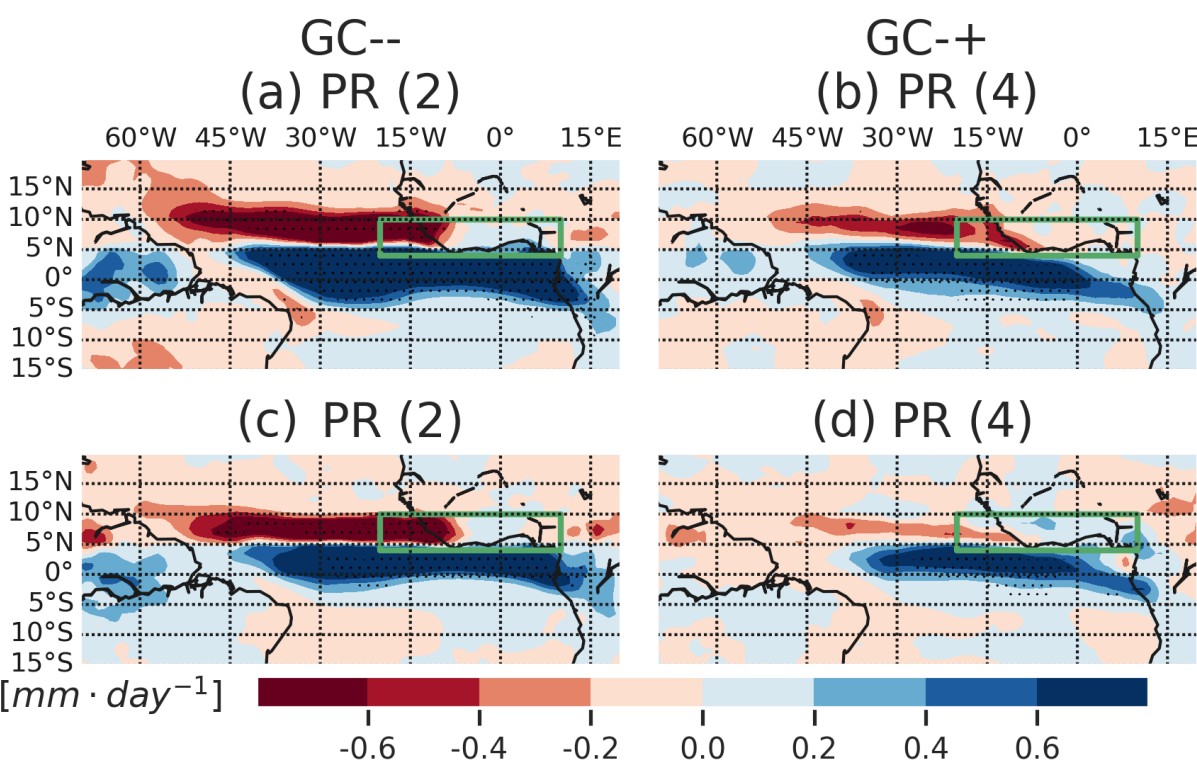

**Figure A10.** 1985-2014 (a)-(b) and 2070-2099 (c)-(d) regression maps of the rainfall anomalies onto the standardized ATL3 index for the GC- - (left column) and GC-+ (right column) groups. Stippling indicates areas where the regression coefficients are significant at 95 % confidence level for at least 50 % of the models in each group, and where more than 80 % of the models agree on the sign of the regression coefficient. The number of models in each group is indicated in parentheses.





## A1   Appendix tables

**Table A1.** Models in the different categories for the 2015-2039 period.

| GC++ | GC+- | GC-+ | GC- - | OC++ | OC+- |
|------|------|------|-------|------|------|
| CAMS-CSM1-0 | ACCESS-ESM1-5 | CNRM-CM6-1 | ACCESS-CM2 | BCC-CSM2-MR | ACCESS-CM2 |
| CanESM5-CanOE | BCC-CSM2-MR | INM-CM4-8 | HadGEM3-GC31-LL | CAMS-CSM1-0 | ACCESS-ESM1-5 |
| EC-Earth3 | CESM2 | KACE-1-0-G | INM-CM5-0 | CESM2 | CESM2-WACCM |
| EC-Earth3-Veg | CESM2-WACCM | | | CNRM-CM6-1 | CNRM-CM6-1-HR |
| FGOALS-f3-L | CNRM-CM6-1-HR | | | CNRM-ESM2-1 | CanESM5-CanOE |
| GFDL-ESM4 | CNRM-ESM2-1 | | | CanESM5 | FIO-ESM-2-0 |
| MCM-UA-1-0 | CanESM5 | | | EC-Earth3 | GFDL-ESM4 |
| MPI-ESM1-2-HR | FIO-ESM-2-0 | | | EC-Earth3-Veg | |
| MRI-ESM2-0 | IPSL-CM6A-LR | | | FGOALS-f3-L | INM-CM4-8 |
| NorESM2-LM | MIROC-ES2L | | | HadGEM3-GC31-LL | INM-CM5-0 |
| UKESM1-0-LL | MIROC6 | | | MCM-UA-1-0 | IPSL-CM6A-LR |
| | MPI-ESM1-2-LR | | | MIROC6 | KACE-1-0-G |
| | NorESM2-MM | | | NorESM2-LM | MIROC-ES2L |
| | | | | UKESM1-0-LL | MPI-ESM1-2-HR |
| | | | | | MPI-ESM1-2-LR |
| | | | | | MRI-ESM2-0 |
| | | | | | NorESM2-MM |



**Table A2.** Models in the different categories for the 2040-2069 period.

| GC++ | GC+- | GC-+ | GC- - | OC++ | OC+- |
|---|---|---|---|---|---|
| CanESM5 | ACCESS-ESM1-5 | CNRM-CM6-1 | ACCESS-CM2 | CAMS-CSM1-0 | ACCESS-CM2 |
| EC-Earth3 | BCC-CSM2-MR | HadGEM3-GC31-LL | KACE-1-0-G | CESM2 | ACCESS-ESM1-5 |
| EC-Earth3-Veg | CAMS-CSM1-0 | INM-CM4-8 | | CNRM-CM6-1 | BCC-CSM2-MR |
| IPSL-CM6A-LR | CESM2 | INM-CM5-0 | | CNRM-ESM2-1 | CESM2-WACCM |
| MRI-ESM2-0 | CESM2-WACCM | | | EC-Earth3-Veg | CNRM-CM6-1-HR |
| | CNRM-CM6-1-HR | | | MCM-UA-1-0 | CanESM5 |
| | CNRM-ESM2-1 | | | MRI-ESM2-0 | CanESM5-CanOE |
| | CanESM5-CanOE | | | | EC-Earth3 |
| | FGOALS-f3-L | | | | FGOALS-f3-L |
| | FIO-ESM-2-0 | | | | FIO-ESM-2-0 |
| | GFDL-ESM4 | | | | GFDL-ESM4 |
| | MCM-UA-1-0 | | | | HadGEM3-GC31-LL |
| | MIROC-ES2L | | | | INM-CM4-8 |
| | MIROC6 | | | | INM-CM5-0 |
| | MPI-ESM1-2-HR | | | | IPSL-CM6A-LR |
| | MPI-ESM1-2-LR | | | | KACE-1-0-G |
| | NorESM2-LM | | | | MIROC-ES2L |
| | NorESM2-MM | | | | MIROC6 |
| | UKESM1-0-LL | | | | MPI-ESM1-2-HR |
| | | | | | MPI-ESM1-2-LR |
| | | | | | NorESM2-LM |
| | | | | | NorESM2-MM |
| | | | | | UKESM1-0-LL |



**Table A3.** Models in the different categories for the 2070-2099 period.

| GC++ | GC+- | GC-+ | GC- - | OC++ | OC+- |
|------|------|------|-------|------|------|
| ACCESS-ESM1-5 | CAMS-CSM1-0 | CNRM-CM6-1 | ACCESS-CM2 | BCC-CSM2-MR | ACCESS-CM2 |
| BCC-CSM2-MR | CESM2 | HadGEM3-GC31-LL | KACE-1-0-G | CNRM-CM6-1 | ACCESS-ESM1-5 |
| CanESM5 | CESM2-WACCM | INM-CM4-8 | | CanESM5 | CAMS-CSM1-0 |
| CanESM5-CanOE | CNRM-CM6-1-HR | INM-CM5-0 | | INM-CM4-8 | CESM2 |
| EC-Earth3 | CNRM-ESM2-1 | | | KACE-1-0-G | CESM2-WACCM |
| EC-Earth3-Veg | FGOALS-f3-L | | | MCM-UA-1-0 | CNRM-CM6-1-HR |
| IPSL-CM6A-LR | FIO-ESM-2-0 | | | MIROC6 | CNRM-ESM2-1 |
| MCM-UA-1-0 | GFDL-ESM4 | | | MRI-ESM2-0 | CanESM5-CanOE |
| MRI-ESM2-0 | MIROC-ES2L | | | NorESM2-MM | EC-Earth3 |
| | MIROC6 | | | | EC-Earth3-Veg |
| | MPI-ESM1-2-HR | | | | FGOALS-f3-L |
| | MPI-ESM1-2-LR | | | | FIO-ESM-2-0 |
| | NorESM2-LM | | | | GFDL-ESM4 |
| | NorESM2-MM | | | | |
| | UKESM1-0-LL | | | | HadGEM3-GC31-LL |
| | | | | | INM-CM5-0 |
| | | | | | IPSL-CM6A-LR |
| | | | | | MIROC-ES2L |
| | | | | | MPI-ESM1-2-HR |
| | | | | | MPI-ESM1-2-LR |
| | | | | | NorESM2-LM |
| | | | | | UKESM1-0-LL |