# Peer review of "Weakened impact of the Atlantic Niño on the future equatorial Atlantic and Guinean Coast rainfall"

_Earth System Dynamics, 2021_

## Author Comment (AC4)

**Answers to the Reviewer 4**

We would like to thank the reviewer for the evaluation of our study, the comments and suggestions. Please, find below our response to your comments, on a point by point basis. Your comments are recalled in red and our responses are written in purple.

**RC4: 'Comment on esd-2021-46', Anonymous Referee #4, 20 Sep 2021**

**General comments:**

This paper investigates the present and future characteristics of the Atlantic Niño and its influence in the Gulf of Guinea and Equatorial Atlantic precipitation, using CMIP6 simulations. It shows that models project a weakening in the variability of the ATL3 region which would then lead to a weakening in the rainfall variability in the equatorial Atlantic and Guinean Coast.

In my opinion the paper has valuable information that is within the scope of ESD and deserves publication after the some revisions.

The paper has a lot of information that not always is presented in the clearest way. Specially when the authors divide the 31 models in different groups (GC+, GC-, OC+). I fell that the group OC+ is not at all necessary in section 4, and makes the discussion of the results a bit messy. Also section 5 is hard to follow in some places.

Thank you for this general comment. We will follow the suggestion and reduce the number of groups of models in the revised version as detailed below.

**Specific comments:**

**Introduction:**

**ATL3 is an index that reflects the variability of the Atlantic Niño region, but I don't think it is correct to use ATL3 acronym to refer to the Atlantic Niño.**

We will keep ATL3 acronym for the Atlantic Niño region. The Atlantic equatorial mode acronym (AEM) will be used instead, to refer to the Atlantic Niño.

**Data and methods:**

**Why do you perform a quadratic detrend of the data?**

**JAS ATL3 index: anomalies, linear and quadratic trends for the 1985-2014 period**

---

## Author Response (AR1)

**Answers to the Reviewer 1**

**RC1**: 'Comment on esd-2021-46', Anonymous Referee #1, 14 Sep 2021

In my view, the paper is within the scope of ESD and presents some interesting new findings, however, there are some questions that I would like to authors to address to improve the presentation.

Abstract: I feel that it is too long and would suggest that you shorten it. For instance, discussions of the Saharan Low can go and the discussions of the Bjerknes feedback more focused.

We will shorten the abstract by removing the discussion about the Saharan Low

L.20: I don't understand how upper-level subsidence leads to the little dry season.

A rapid shift of the rainfall belt from the coastal regions of West-Africa to the Sahel occurs around the 21$^{st}$ of June, and marks the onset of the monsoon season. In July and August, the development of the Atlantic cold tongue is at its maximum (due to upwelling over the ATL3 area), which leads to higher pressure over the Gulf of Guinea and the coastal regions of West Africa. As a consequence, during July and August, the atmospheric circulation above Guinea Coast is characterized by a downward flow at upper levels which reduces air ascent over Guinea Coast and leads to the so call "little dry season" (see also section 2.1.6.2 The Little Dry Season of the book: Meteorology of Tropical West Africa, The Forecasters' Handbook, 2017). This section will be rewritten accordingly.

L.25: Does the little dry season occur in AMJ? One should expect this to correspond to the period between the double rainfall peaks?

The little dry season occurs every year in July and August, and indeed it corresponds to the period between the double rainfall peaks.

L.40: Jouanno et al 2017 is just one of the numerous papers that discussed the Bjerknes feedback, so I don't understand why this paper is presented as the final say. I think the point is that several processes drive the Atlantic Niño variability. The next paragraph, the upper-level links to the Indian and Pacific Oceans need to be better explained.

We agree that many papers discussed the Bjerknes feedback. There are also many studies pointing out different forcings on the development of an Atlantic Niño event, as you stated and this is mentioned in the paragraph including this sentence. In this last sentence, we would like to present the paper of Jouanno et al 2017 as an example of a study which disentangled the relative contributions of ocean dynamics and the thermodynamic processes in the control of the Atlantic Niño/Niña development (and not as a final say). We propose to rewrite this last sentence of the line 40 to clarify this point:

" *In order to quantify the relative contributions of the different processes driving the Atlantic Niño variability, Jouanno et al 2017 highlighted the dominant role of the dynamical forcing (i.e the Bjerknes feedback) relative to the thermodynamic processes (i.e air-sea heat flux exchanges). They argued that biases in the atmospheric components of most of the GCMs participating in the CMIP project lead to the underestimation of the dynamic part of the Atlantic Niño forcings*".

Second point: improvement of the upper-level links to the Indian and Pacific Oceans.

We propose to add additional information:

*The general response of the atmosphere to Atlantic Niño positive phases is a modification of the Walker circulation, characterized by rising motion and upper-level divergence in the Atlantic region and compensating upper-level convergence and sinking motion in the central Pacific that also triggers a Gill-type response in vorticity. The Gill-type response is characterized by a pair of upper-level anticyclones to the west and a pair of upper-level cyclones to the east of the abnormal warm oceanic regions. These upper-level divergence and vorticity responses are related to each other by Sverdrup balance (Hamouda and Kucharski, 2019, Losada et al., 2010, Kucharski et al., 2009). The divergence and vorticity responses are generally baroclinic, and are of opposite sign at low levels, meaning in the Indian region a low-level anticyclone is present that leads to reduced Indian Monsoon rainfall (Kucharski et al. 2007, 2008, 2009). On the other hand, the sinking motion in the central Pacific can lead to easterly surface winds in the central-western Pacific that could potentially lead to a La-Niña event (Rodriguez-Fonseca et al, 2009). Finally, the development of a La-Niña event due to a warm phase of the Atlantic Niño would favor positive rainfall anomalies over the Indian Peninsula, which would counteract the negative rainfall anomalies associated with the Atlantic Niño (Ding at al, 2012).* "

Ding, H., Keenlyside, N.S. & Latif, M. Impact of the Equatorial Atlantic on the El Niño Southern Oscillation. *Clim Dyn* **38,** 1965–1972 (2012). https://doi.org/10.1007/s00382-011-1097-y

Hamouda, M. E., and F. Kucharski, 2019: Ekman pumping mecha- nism driving precipitation anomalies in response to equatorial heating. Climate Dyn., 52, 697–711, https://doi.org/10.1007/s00382-018-4169-4.

Kucharski F, Bracco A, Yoo JH, Tompkins A, Feudale L, Ruti P, Dell'Aquila A (2009) A Gill-Matsun-type mechanism explains the Tropical Atlantic influence on African and Indian Monsoon rainfall. Quart J R Met Soc 135:569–579

Kucharski F, Bracco A, Yoo JH, Molteni F (2007) Low-Frequency variability of the Indian monsoon–ENSO relationship and the tropical Atlantic: the ''weakening'' of the 1980s and 1990s. J Clim 20:4255–4266

Kucharski F, Bracco A, Yoo JH, Molteni F (2008) Atlantic forced component of the Indian monsoon interannual variability. Geophys Res Lett 35. doi:10.1029/2007GL033037

Losada T, Rodríguez-Fonseca B, Polo I, Janicot S, Gervois S, Chau- vin F, Ruti P (2010) Tropical response to the Atlantic equatorial mode: AGCM multimodel approach. Clim Dyn 35(1):45–52. https ://doi.org/10.1007/s0038 2-009-0624-6

Rodríguez-Fonseca, B., Polo, I., García-Serrano, J., Losada, T., Mohino, E., Mechoso, C. R., and Kucharski, F. (2009), Are Atlantic Niños enhancing Pacific ENSO events in recent decades? *Geophys. Res. Lett.*, 36, L20705, doi:10.1029/2009GL040048.

L.50: What destructive interference means or how it operates is not clear at all.

We propose to rewrite this point.

*"Considering the tropical basins separately, an anomalous warming of the eastern equatorial Atlantic induces a dipolar rainfall response over West-Africa in boreal summer: a decrease of the rainfall in the Sahel region and an increase of the rainfall over Guinea Coast. However, below normal sea surface temperatures in the eastern tropical Pacific lead to an increase of the rainfall in the Sahel. After 1970s, the coupling between the eastern equatorial Atlantic and the eastern tropical Pacific has strengthened, and the two basins are characterized by an opposite phase relationship. Therefore, a positive phase of the Atlantic Niño is associated with negative SST anomalies in the eastern tropical Pacific. This leads to rainfall anomalies of opposite signs over the Sahel, which damps the West-African dipolar rainfall response associated with the Atlantic Niño (Losada et al 2012)"*

Losada T, Rodriguez-Fonseca B, Mohino E, Bader J, Janicot S, Mechoso CR (2012) Tropical SST and Sahel rainfall: a non-stationary relationship. Geophys Res Lett. https ://doi. org/10.1029/2012g l0524 23

L.55: I don't think that the discussion "...temperature and precipitation over the globe" is necessary here. I'd suggest that you remove that and keep the flow focused on the equatorial Atlantic.

This discussion is removed as suggested.

*"Results from the General Circulation Models (GCMs) participating in the Coupled Model Intercomparison Project (CMIP) show that from the fifth phase (CMIP5) to the sixth phase (CMIP6) of the Coupled Model Intercomparison Project, the surface temperature biases have been reduced over the tropical Atlantic, as pointed out by Richter and Tokinaga (2020) in an analysis of the pre-industrial control experiment performed with 33 models."*

L.100: Is it "realistic" or observed natural and anthropogenic forcing?

CMIP6 historical simulations are forced with observed natural and anthropogenic forcings. We will modify the text accordingly.

L.105: "These latter simulations…" Do you mean SSP-85?

Yes, we mean SSP5-85, we will modify the text accordingly.

L.115: Please use one rainfall, SST etc data to compare with the models. Comparing multiple observations is unnecessary and it makes following your discussions difficult.

We will keep ERA5 reanalysis for comparison with the model outputs as our main conclusions do not depend on the choice of the selected observation product. The other observed rainfall and SST datasets will be removed.

L.130: Why do you use quadratic detrending, are the trends quadratic? I ask because we are more used to linear trends. More explanation is needed here.

[Figure]

*Figure R 1. SST indices of the Atlantic Niño: JAS mean of monthly SST anomalies averaged over the Atlantic Niño area, for the 1985-2014 period (green curves). The linear (blue curves) and quadratic (orange curves) trends are superimposed on each panel. SST outputs from CMIP6 historical simulations (30 GCMs) and the ERA5 reanalysis are considered.*

From Fig. R1, we noted that the quadratic trend does not differ from the linear in much of the models (e.g., ACCESS-CM2, MRI-ESM2-0, NORESM2-LM). However, there are some cases where both trends behave differently, e.g. EC-Earth3-Veg, GFDL-ESM4. This motivated us to consider the quadratic trend which, would better follow the changes in the trends inside each time series.

[Figure]

*Figure R 2 Residuals of the detrended JAS ATL3 index after removing the linear trend (blue curves) and the quadratic trend (orange curves). The displayed 1985-2014 time series are from 30 CMIP6 models and ERA5.*

The residuals from the linearly detrended SST time series are considered in Fig. R2. Results show that there is no substantial difference between the residuals when the linear or the quadratic trends are removed. Therefore, for simplicity, we will consider detrending linearly the different datasets in the revised manuscript.

Table 2: I don't find the numerous acronyms here very useful and they can as well cause more confusion, given that we have the model names to deal with. Atl3 is widely known as the SST anomalies (well, or some other quantities) averaged in the Atlantic Niño region defined as 0-20W, 3N-3S; so it's not necessary to introduce a new definition ATL3B. Why define TAB1, TAB2 when there are well known regions like Atl4, tropical North Atlantic (TNA) and tropical South Atlantic (TSA)?

We will remove the acronym ATL3B, and keep ATL3 describing the box related to the Atlantic Niño center of action.

TAB1 and TAB2 are two domains used in our work to validate the SST and rainfall patterns, respectively, related to the Atlantic Niño mode. In this validation process, we wanted to consider a large spatial domain that takes into account both the northern and southern parts of the tropical Atlantic, which is not the case for the TNA and TSA regions. The ATL4 region considers only the western part of the equatorial Atlantic, which is too narrow relative to our objective.

We will add more argument in the description of the domains in the revised version of the manuscript.

Data and methods section: For easy navigation, I would suggest splitting this section, for instance "Data", "CMIP6 Models", "Analysis strategy" or something similar

We would like to thank you for this suggestion. This section will be reorganized like this:

2.   Data and methods

   2.1. CMIP6 data
   2.2. Reanalysis
   2.3. Analysis strategy

section 3: I don't find the line plots and discussions of bimodal structure and annual cycle necessary and I suggest that you remove these. I consider the question of annual cycles and seasonality as a separate question. Since this study is about JAS, it is enough to briefly describe the JAS patterns and biases and move to the Atlantic Niño related SST and rainfall and their future changes.

The question of annual cycles and seasonality will be moved to the supplementary material, and we will focus the discussion on the seasonal biases and patterns as suggested.

Fig. 2: How do you have strong easterly wind biases over warm SST biases? This pattern needs explanation because it is inconsistent with expectation and inconsistent with Richter and Tokinaga 2020 (see their Fig. 2).

[Figure]

*Figure R 3 Ensemble mean of the JAS SST (in colors), rainfall (in contours) and 10 wind (arrows) biases relative to ERA5 over 1985-2014. Theses biases are computed from 23 GCMs for which this field is available.*

The strong easterlies shown in our figure (north of 10°S in the eastern basin) are part of the anomalous northerly flow that brings the moist air into Guinea Coast, favoring a positive rainfall bias. This inflow is strong at 850 hPa (roughly 1.5 km above the sea level and this is the reason we selected this level on Figure 2.). The pattern is different from the one of the bias in the near-surface flow (10 m) that is shown in Richter and Tokinaga 2020.

In Figure R3, we show the anomalous near-surface westerly biases for JAS, which is consistent with Richter and Tokinaga 2020. This figure will be added to the supplementary material and briefly discussed.

Section 4: You are basically evaluating SST and rainfall patterns rather than "teleconnections". Secondly, regression maps show rainfall, SST etc with units in mm/day, degC etc and I don't see the need for the repeated use of regression coefficient, instead of referring to rainfall, SST etc.

We will refer directly to the variables (rainfall, SST, etc), by removing the regression coefficient terms.

L.225: Again, I don't see the need to compare observations with other observations here. I suggest that the authors rather use just one observational data to compare the CMIP models.

We have decided to keep only the ERA5 reanalysis to compare with the CMIP6 models.

Fig. 4: I suggest separating this Figure so that the maps stand as one Figure, and the Taylor diagram stands as a different Figure. In the Taylor diagram, the REF which is here ERA5 should correspond to a standard deviation of 1. The authors should explain why/how their scaling leads to a different value. Again I suggest that the authors discuss the overall model fidelity using both pattern correlations and variance (that is closeness to REF).

This figure will be split into two, as suggested.

The standard deviation of the SST spatial patterns related to the Atlantic Niño in the different models is not scaled by its corresponding value in ERA5. This is why the standard deviation of the reference is different from 1 in our Figure (4b). Scaling the data will not change the figure, and we chose to have the estimation of the deviation from the spatial mean of the pattern. We will add this additional information to the title of the figure. Overall, the models show a good representation of the SST spatial distribution associated with the Atlantic Niño, with an overestimation of the SST amplitude.

Fig. 5: Again I suggest two different figures: one for the maps and the other for the Taylor diagram. One satellite rainfall data and one SST data should do, no need to compare different observations which I consider outside the scope of this manuscript.

We will split the figure into two, as suggested, and we will consider ERA5 reanalysis for the model evaluations.

L.260: It'll be good to state what sea surface heights represent, what understanding you'll like to gain by analyzing that. The same could be said of the atmospheric variables. The motivation and physical reasoning behind the analysis need to be better formulation. For instance, SSH-→SST(atl3) regression implying thermocline impact on the SSTs which is one element of the

Bjerknes feedback (Keenlyside and Latif, 2007). Then the winds/SST regressions another element?

As suggested, the reasoning below the analyses will be explained in the revised version. During positive phases of the Atlantic Niño, warmer than normal sea surface in the eastern equatorial Atlantic weakens the zonal surface pressure gradient, which in turn weakens the prevailing trade winds. The regression of the low-level zonal component of the wind onto the ATL3 index is used to evaluate the first component of the Bjerknes feedback, which is the forcing of the surface wind in the west basin of the Atlantic Ocean by SST in the eastern basin. Then, these anomalous westerlies increase the surface convergence above the warm waters in the east, which leads to a rising of the sea surface height, an increased heat content and a deepening of the thermocline. This is the second component of the Bjerknes feedback. Then, the deepening of the thermocline reduces the influence of the upwelling of cold subsurface water on the surface temperature, which then reinforces the initial surface warming. This is the third component of the Bjerknes feedback, which we accessed by regressing the sea surface height, a proxy for the thermocline depth, onto the ATL3 index.

Are the rainfall and divergence related to the ITCZ/atmospheric component of the Bjerknes feedback (Nnamchi et al. 2021)?

Yes, the rainfall and divergence are related to the atmospheric ITCZ component of the Bjerknes feedback. This is related to the spurious southward position of the mean ITCZ position in the climate models relative to the observations during the boreal summer. This bias would lead to an enhancement of the coupling between the atmosphere and the ocean, during the growing phase of the Atlantic Niño in the models. We will add a short discussion of this point inthe revised manuscript.

L.310: How you calculated the percentages should be explained in context here so that it's understood what minus percentages, plus percentages mean.

The percentage of change of the ATL3 index standard deviation between two periods is computed as $100 \times \frac{\sigma_{fut} - \sigma_{his}}{\sigma_{his}}$, where $\sigma_{his}$ is the standard deviation of the JAS ATL3 index in the 1985-2014 period, and $\sigma_{fut}$ the standard deviation of the JAS ATL3 index in a future period (the near-term, mid-term or long-term periods). This information will be added to the revised version of the article.

L.315: This point needs more discussions/explanations of why your result is different from Brierley and Wainer (2018), Is it because the use different time slices, methods, models etc?

There could be several reasons to explain the differences between our results and those from Brierley and Wainer (2018). Among others, we can postulate on the differences in the models between the CMIP5 and CMIP6. Brierley and Wainer (2018) compared a 1% per year quadrupled $CO_2$ experiment to a pre-industrial control simulation of CMIP5, which is different from the simulations compared in our analysis (historical and SSP5-8.5 simulations). A better comparison between the two studies could be performed by analyzing the ATL3 variability changes between CMIP6 1pctCO2 and the CMIP6 pre-industrial simulations. We will add this comment to the revised manuscript.

L.360: It's important to first outline the elements of the Bjerknes feedback as the basis for your analysis and then build the subsequent discussions around that.

We will recall the three elements of the Bjerknes feedback at this point.

L.370: I think that this paper is about weakening equatorial Atlantic variability rather than teleconnections. Please note that equatorial Atlantic doesn't mean the same thing as tropical Atlantic.

We will rewrite this point, by replacing the tropical Atlantic rainfall teleconnection by the equatorial Atlantic rainfall variability. Thank you for this correction.

Fig. 11: I don't really find the discussions of Saharan Low interesting at all because I feel the equatorial region is enough to interpret the results here. The Sahara/Sahel matter is a different topic.

We will remove the discussion about the Saharan Low and keep only the link between the mean state change along the equatorial Atlantic and the change of the Atlantic Niño variability.

**References**

Brierley, C. and Wainer, I.: Inter-annual variability in the tropical Atlantic from the Last Glacial Maximum into future climate projections simulated by CMIP5/PMIP3, Climate of the Past, 14, 1377–1390, https://doi.org/10.5194/cp-14-1377-2018, 2018.

Jouanno, J., Hernandez, O., and Sanchez-Gomez, E.: Equatorial Atlantic interannual variability and its relation to dynamic and thermodynamic

processes, Earth System Dynamics, 8, 1061–1069, https://doi.org/10.5194/esd-8-1061-2017, 2017.

Keenlyside, N. S., & Latif, M. (2007). Understanding Equatorial Atlantic Interannual Variability, Journal of Climate, 20(1), 131-142. https://journals.ametsoc.org/view/journals/clim/20/1/jcli3992.1.xml

Nnamchi, H.C., Latif, M., Keenlyside, N.S. et al. Diabatic heating governs the seasonality of the Atlantic Niño. Nat Commun **12,** 376 (2021). https://doi.org/10.1038/s41467-020-20452-1

Richter, I., Tokinaga, H. An overview of the performance of CMIP6 models in the tropical Atlantic: mean state, variability, and remote impacts. Clim Dyn **55,** 2579–2601 (2020). https://doi.org/10.1007/s00382-020-05409-w

**RC2**: 'Comment on esd-2021-46', Anonymous Referee #2, 17 Sep 2021  reply
**Review of the manuscript ESD-2021-46, "Weakened impact of the Atlantic Niño on the future equatorial Atlantic and Guinean Coast rainfall", by Koffi Worou, Hugues Goosse, Thierry Fichefet, and Fred Kusharski.**

The submitted manuscript explores the rainfall annual cycle in the Guinea Coast, gives a detailed analysis of the future changes in the Atlantic Niño and their impact on the rainfall, and the modulation of the Bjerknes feedback in the future climate change projection. The investigation is based in 31 historical simulations from General Circulation Models of CMIP6 with some observations and reanalysis. The authors found that these models are able to simulate reasonably well the rainfall annual cycle in the Guinea Coast with a wet bias in boreal summer (July-August-September). They also found a rainfall decrease in the Tropical Atlantic region due to a weakening of the Bjerknes feedback over the equatorial Atlantic in future climate projection. This work will be a valuable contribution to Earth System Dynamics journal after some revisions.

The paper is well written, well documented, easy to follow and understand from the beginning up to section 4. In section 5, there are lots of information, and it's a bit dense. Please, what is the purpose of defining all the groups you defined? I am referring to group GC+, GC-, GC+-, GC++, OC+, OC-, ect…

By defining these different groups, we aim to understand if different group of models simulate the rainfall pattern related to the Atlantic Niño over the equatorial Atlantic and the Guinea Coast in different ways and in particular if a different simulation of the current state has some implications on the simulated future changes in rainfall patterns. We also aimed to highlight the differences in the key physical mechanisms between the groups. Focusing on the Guinea Coast for example, we first identify the climate models which are able to simulate realistically the observed rainfall pattern related to the Atlantic Niño in the Guinea Coast over the past decades (the group GC+). In observations, a positive rainfall anomaly over Guinea Coast is related to a warm phase of the Atlantic Niño (and vice versa). The group GC- indicates models which presents a negative rainfall pattern associated with a warm phase of the Atlantic Niño. For the future changes in the ATL3-related rainfall pattern over Guinea Coast, we separated models which present and enhancement of the positive rainfall pattern (GC++) from models which present a weakening of the rainfall pattern (GC+-). Then we tried to understand the reasons of these changes and the differences between the different categories. A similar argument is applied to the ocean. However, in the revised version of the paper, we will focus our analyses on the Guinea Coast rainfall changes related to the Atlantic Niño. We will reduce the number of groups to three: GC++, GC+-, and the multi-model ensemble mean (EnsMean)

It will be nice if you could resume your findings about the previous mentioned groups in a table.  Moreover, it is better to name the figure you are referring too early in the text than in the middle or at the end of a paragraph.

We will resume our findings about the different groups in a table, as suggested.

We will also name the figure early as suggested.

**Minor**

Line 35: "deepens" not "deepen"

Thank you for this correction. It will be taken into account in the revised manuscript.

Please, add a figure in the supplement material to show the different boxes of Table 2.

The different boxes will be added in the Figure 2 of the first version of the manuscript, as suggested by the reviewers 4 and 5. If needed, we will add a new figure in the supplementary material as suggested.

Provide a statement on how the data used for the study could be accessed.

The CMIP6 data and the reanalysis ERA5 will be used in the revised manuscript. We will specify how to retrieve these datasets.

Line 68: Precise the figure you refer to after giving the interval of the RMSE.

We were referring to the figure (1a). However, this figure will be put in the supplementary material in the revised version of the paper.

Line 160 and line 193: Which figures are you referring to? If it is not shown, please precise.

We were referring to the figure (1a), which will be put in the supplementary material in the revised manuscript.

Line 197: Why you did not represent the bias relative to ERA5 instead of the mean state?

We did not represent the bias relative to ERA5 for each model because we already showed the multi-model ensemble mean of the model biases in Fig. 2.

Line 216: The multimodel mean "underestimates" the SST STD in relation to ERA5 in the time period you have highlighted. Compared to other observations, the multimodel underestimates the SST STD in May-June and overestimates it the rest of the year.

Thank you for this correction, we will take it into account in our revision.

L256-259: There are 24 GC+ and 6 GC- models. It seems like 1 model is missing because there are 31 GCMs in total.

Yes, the model GISS-E2-1-G is discarded, because it has no significant sign-dependent average of the rainfall anomalies related to the Atlantic Niño over Guinea Coast. However, in the revised manuscript, we will discard this model, so the total number of models will be 30.

Title of figure 6d: It is better to write ERA5 (ORAS5) than ERA5/ORAS5, or use only ORAS5, because it is confusing.

We will use ERA5 (ORAS5) in the title, thank you for the suggestion.

L260: Please, precise "not shown" after "nor in the observations".

It was shown in Fig. (5a) on the CMAP-HADISST and GPCP-ERSST maps. However, we will remove these maps in the revised manuscript, and will precise "not shown" as suggested. Thank you.

L282: Please, precise the figure you are referring too at the end of the sentence (Fig. A4?).

Yes, we are referring to Fig. A4. We will precise this figure at the end of the sentence as suggested. In addition, this section will be profoundly modified in the revised manuscript, as suggested by one reviewer.

Caption of figure A4, add the color of the box for each region.

This information will be added in the revised manuscript.

L295: "The models show a poor to modest spatial correlation with ERA5, which ranges from −0.4 to 0.6". Precise the figure you are referring to, is it (Fig. 5a?)

We are referring to the Fig. (5b).

L298: Is OC+ the sum of GC+ and GC- when referring to EAB region?

Yes, OC+ is the sum of GC+ and GC-.

You did not use Figure A6. Please, remove the figure if it is not needed.

Figure A6 will be removed, as it is not needed. Thank you for the suggestion.

Figure 7b: Please, keep the same color in (a) and (b) for the period 2015-2039, 2040-2069, and 2070-2099.

The same colors will be kept for the same periods as suggested.

L320: Refer the figure after 0.32°C.

We will refer to the Fig. (8a) in the revised manuscript.

L397: Remove one "zonal".

Thank you for this correction, it will be removed.

**RC3**: 'Comment on esd-2021-46', Anonymous Referee #3, 17 Sep 2021  reply

1. **General Comments**

The authors of this manuscript investigate the present-day and future boreal summer (JAS) rainfall and sea surface temperature (SST) variability in the eastern and equatorial Atlantic using 31 historical and scenario simulations from the sixth phase of the Coupled model Intercomparison Project (CMIP6). They show that the rainfall annual cycle, computed for the period 1985-2014, in the Guinea Coast is generally well simulated. Yet, a wet bias persists in boreal summer due to a large SST bias in the eastern equatorial Atlantic and south Atlantic regions. The rainfall variability is strongly linked to the SST variability in this region and therefore the SST variability in the eastern equatorial Atlantic is also investigated. The authors show that relative to the present-day situation, in a climate with a high anthropogenic emission of greenhouse gases, the eastern equatorial Atlantic JAS SST variability weakens. They show that the reduced SST variability in the equatorial Atlantic could be due to a weakening of the Bjerknes feedback. As a result, relative to the present-day situation, in the future they also find a reduction of the rainfall variability over the equatorial Atlantic Ocean and Guinea coast in a majority of the CMIP6 considered.

The article is well written and addresses an important topic with detailed results. In my view the results are within the scope of ESD and therefore, I recommend minor revision before publication following the different aspects provided bellow.

2. **Specific Comments**

**Abstract**

**L3**. I would state that both historical and scenario (SSP5-8.5) simulations from 31 GCMs from CMIP6 are used throughout the study.

We will modify the statement as suggested, thank you.

**L6**. Add "boreal" to the sentence "This bias is associated with too high mean summer SSTs"

 The boreal summer will be explicitly mentioned in the revised manuscript.

**Introduction**

**L31**. The acronym ATL3 is used in this study to refer to the Atlantic Niño. It is generally more used to define the region where the Atlantic Niños occur (20°W-0°; 3°S-3°N). The authors have defined this region with the acronym ATL3B in Table 2 but ATL3B is never used in the manuscript.

We will remove the acronym ATL3B, and keep ATL3 for the Atlantic Niño region.

**Data and methods**

**L132**. Please, can you explain how the anomalies were computed. Do you remove the climatological monthly-mean seasonal cycle?

The climatological monthly mean is first removed from each data set, for each considered period. The resulting anomalies are then quadratically detrended and averaged over three months, July-August-September. In the revised paper, we will linearly detrend each monthly anomalies of each data before averaging over the JAS season. We will add this information in the revised manuscript.

**Section 3.2**

**L200**. The title of the section says "JAS mean" but the JAS mean is not discussed.

We missed this discussion. It will be taken into account in the revised version of the paper. Thank you for this remark.

**L214**. "The winter Atlantic Niño" was defined as the "Atlantic Niño II" by Okumara and Xie (2006).

We will refer to this article in the revised manuscript.

Okumura, Y., & Xie, X. (2006). Some overlooked features of tropical atlantic climate leading to a new Nino-like phenomenon. Journal Of Climate, 19, 5859-5874. doi:10.1175/JCLI3928.1

**Figures 1 and 3:**

I would like to see the cross-correlation between the GC rainfall anomalies and ATL3 SST anomalies. As in the CMIP6 ensemble, both the ATL3 SST variability and the GC rainfall variability peak in JJA and assuming that the maximum of correlation is found at 0 lag then why not regressing JJA SST (rainfall) anomalies on the standardized JJA ATL3 index in Figure 4 (Figure 5)?

[Figure]

*Figure R 4 Monthly rainfall anomalies of Guinea Coast regressed onto the JJA (a) and JAS (b) standardized ATL3 index over the 1985-2014 period. Outputs from 30 CMIP6 historical simulations and ERA5 are analyzed. Gray vertical bands indicate the SST season considered in each case.*

The Figure R4 shows the monthly stratified regression of the Guinean Coast rainfall onto the JJA and JAS standardized ATL3 index over the 1985-2014 period. Results indicate that the CMIP6 ensemble mean response of the Guinean Coast rainfall is maximum over the JAS season in both cases. As our study is focused on the impact of the Atlantic Niño on the Guinea Coast rainfall, we therefore consider the JAS season instead of the JJA and we will explain this choice in the revised manuscript

**Section 4.1:**

To investigate the boreal summer Atlantic Niño pattern why not regressing the JJA SST anomalies onto the standardized JJA ATL3 index?

As shown and discussed in section 3.2 the ATL3 variability peaks in JJA in the CMIP6 ensemble corresponding to the Atlantic Niño activity in the ATL3 region. Therefore, when looking at future Atlantic Niño changes, I would recommend to use JJA and not JAS.

We are interested in the covariability between the Atlantic Niño and the rainfall in Guinea Coast, which peaks in JAS in the CMIP6 models, as displayed in Fig. R4. This is our motivation for the choice of this season. However, we verified that our conclusions remain unchanged whether we choose JJA or JAS. For instance, Figure R5 shows a weakening of the SST pattern related to the Atlantic Niño in the future periods relative to 1985-2014 both in JJA and JAS.

[Figure]

*Figure R 5  Regression maps of the SST anomalies onto the standardized ATL3 index for the JJA (a-d) and JAS (e-f) seasons and four different periods. Displayed maps correspond to the multi-model ensemble mean patterns from 30 CMIP6 models. Stippling indicates grid points where more than 50% of the models show significant coefficients at 95% level and more than 80% of the models agree on the sign of the regression coefficient.*

**Figure 4**: One could draw the TAB1 and TAB2 boxes on Figure 4 if it stays legible.

TAB1, TAB2 and ATL3 regions will be drawn on the figure.

**Section 4.2:**

**L241**. Should be: "Figure 5(a) displays the regression maps of the JAS rainfall anomalies onto the standardized JAS ATL3 index", correct?

Yes, the statement is correct. Thank you, we will take it into account in the revised manuscript.

**L254-L256**: 31 models are present on Figures 4, 5 and 6 but only 30 models are in the GC groups (24 + 6).

Yes, this is because the model GISS-E2-1-G has been discarded, as the sign-dependent average of the rainfall anomalies related to the Atlantic Niño is insignificant over Guinea Coast. In the revised manuscript, the GISS-E2-1-G model will not be used, so that the total number of models will be 30. We will add this information to the revised manuscript.

**Figure 6. caption:** Should "associated with the standardized ATL3 index" be "associated with the JAS standardized ATL3 index"?

Yes, the correction will be applied in the revised manuscript.

**Section 5.1:**

**Figure 7:** From (a) to (b) I recommend the authors to keep the same color for the different periods.

The same color will be kept for the same periods.

**Figure 9 caption:** Should "Rainfall anomalies associated with ATL3" be "JAS rainfall anomalies associated with JAS standardized ATL3 index"? Same question for the rest of the subpanels.

Yes, the remark is correct. However, this figure will be removed from the revised manuscript, as suggested by the reviewer 5, as it contains redundant information already depicted in Figure 8.

**L384**. Should "First, the GC+ group (the 24 models in Sect. 4.2 which simulate a realistic GCB rainfall associated with one standard deviation of the ATL3)" be "First, the GC+ group (the 24 models in Sect. 4.2 which simulate a realistic JAS GCB rainfall associated with the standardized JAS ATL3 index)"?

Yes, the remark is correct, we will take it into account, thank you.

**Section 5.2:**

Throughout this section, the authors should state that they investigate the JAS rainfall, SST, 850 hPa zonal wind, moisture flux associated with the JAS standardized ATL3 index.

This comment will be added to the beginning of the section 5.2 in the revised manuscript.

**Figure 10 caption:** Should "Long-term changes of the JAS rainfall (a-e), SST (f-j), 850 hPa zonal wind (k-o), sea surface height (p-t), moisture flux (vectors) and moisture flux divergence (in colors) (u-y) regression patterns associated with the standardized ATL3 index " be "Long-term changes of the JAS rainfall (a-e), SST (f-j), 850 hPa zonal wind (k-o), sea surface height (p-t), moisture flux (vectors) and moisture flux divergence (in colors) (u-y) regression patterns associated with the standardized JAS ATL3 index"?

Yes, the suggestion is correct. It will be taken into account in the revised version of the figure and the manuscript.

**Appendix:**

**Figure A8, A9, A10. caption:** Should "regression maps of rainfall anomalies onto the standardized ATL3 index" be "regression maps of the JAS rainfall anomalies onto the JAS standardized ATL3 index"?

Yes, the season will be added to the standardized ATL3 index in the three figures.

**Figure A6** is not discussed.

This figure will be removed from the revised manuscript. Thank you a lot for your comments.

**Answers to the Reviewer 4**

**RC4**: 'Comment on esd-2021-46', Anonymous Referee #4, 20 Sep 2021

General comments:

 This paper investigates the present and future characteristics of the Atlantic Niño and its influence in the Gulf of Guinea and Equatorial Atlantic precipitation, using CMIP6 simulations. It shows that models project a weakening in the variability of the ATL3 region which would then lead to a weakening in the rainfall variability in the equatorial Atlantic and Guinean Coast.

In my opinion the paper has valuable information that is within the scope of ESD and deserves publication after the some revisions.

The paper has a lot of information that not always is presented in the clearest way. Specially when the authors divide the 31 models in different groups (GC+, GC-, OC+). I fell that the group OC+ is not at all necessary in section 4, and makes the discussion of the results a bit messy. Also section 5 is hard to follow in some places.

 Thank you for this general comment. We will follow the suggestion and reduce the number of groups of models in the revised version as detailed below.

Specific comments:

Introduction:

ATL3 is an index that reflects the variability of the Atlantic Niño region, but I don't think it is correct to use ATL3 acronym to refer to the Atlantic Niño.

We will keep ATL3 acronym for the Atlantic Niño region. The Atlantic equatorial mode acronym (AEM) will be used instead, to refer to the Atlantic Niño.

Data and methods:

Why do you perform a quadratic detrend of the data?

[Figure]

*Figure R 6 Linear (in blue) and quadratic (in orange) trends of the JAS ATL3 index (in green) from 30 CMIP6 models and ERA5. Time series are displayed for the 1985-2014 period.*

We performed the quadratic trend because in the CMIP6 models, the trend of the JAS ATL3 index is not linear all the time (e.g. GFDL-ESM4, Fig. R6). By plotting both linear and quadratic trends, we noticed that when the trend is linear, both trends are similar, whereas in other cases, the quadratic trend departs from the linear trend. However, we now plot the residuals from the detrended SST (Fig. R7) and rainfall data and we note very small differences which do not justify the use of the quadratic trend. Therefore, the revised manuscript will be based on the linearly detrended monthly data (see also comments to the reviewer 1).

[Figure]

*Figure R 7 Residuals of the linearly (blue curves) and quadratically (orange curves) detrended JAS ATL3 index of 30 CMIP6 models and ERA5 over the 1985-2014 period.*

Section 3.1:

Figure 2: I don't understand the wind pattern. It is not consistent with Richter and Tokinaga (2020).

The inconsistency of the wind pattern comes from the difference in the vertical level in Richter and Tokinaga (2020), compared to the level used in our study. We used the 850 hPa horizontal wind (around 1.5 km ) in our study, against the 10 m level in Richter and Tokinaga (2020). Moreover, in Figure R8, we show that the JAS 10m horizontal wind biases are consistent with Richter and Tokinaga (2020). We will add this figure to the supplementary material.

[Figure]

*Figure R 8 Ensemble mean of the JAS SST (in colors), rainfall (in contours) and 10 wind (arrows)  biases relative to ERA5 over 1985-2014. Theses biases are computed from 23 GCMs*

Please add in figure 2 or in any additional figure the boxes defined in table 2.

We will add the boxes in the Figure 2 as suggested.

Section 3.2:

L. 213: I think that the statement "the winter Atlantic Niño has greatly influenced the ENSO events" is a bit too strong.

We will rewrite this sentence, by suppressing the adverb "greatly".

Also you should refer to Okumura and Xie (2006) when talking about the winter Atlantic Niño for the first time.

We will refer to this article when talking about the winter Atlantic Niño for the first time.

Okumura, Y., & Xie, X. (2006). Some overlooked features of tropical atlantic climate leading to a new Nino-like phenomenon. Journal Of Climate, 19, 5859-5874. doi:10.1175/JCLI3928.1

Section 4:

I would reorganize this section. I fell that section 4.1 belongs to section 3, in which the model performance of the patterns are described. Then, in section 4, the authors can focus in the impact of the Atlantic Niño on rainfall.

Thank you for this comment. In the revised manuscript, the section 3 will talk about the seasonal mean rainfall in Guinea Coast (section 3.1) and seasonal SST in the ATL3 region (section 3.2). We

will move the annual cycle discussion to the supplementary material. Then, we will add the model performance on the representation of the Atlantic Niño SST patterns (section 3.3).

The authors jump from figure 5 to figure 6 and back in a confusing way.

We will take this remark into account, by improving the text in the revised version of the manuscript.

I don't think that the analysis of OC+ models is necessary here.

The OC+ group will be removed from the main discussions, thank you for this suggestion.

Section 5:

Again is difficult to follow. I would rearange figures 8 an 9 by areas, to make the discussion in the section easier to follow.

In the revised manuscript, figure 9 will be removed, as suggested by the reviewer 5. This will make the discussion easier to follow. In figure 8, five panels among 6 are about the ATL3 region, while the remaining panel concerns the Guinea Coast region, and we think that there is no other optimal way to rearrange the main panels by area. We will keep this figure and will improve the text.

I would talk about OC+ models only from section 5.3 onwards. I doesn't seem necessary before and makes the discussion hard to follow.

The OC+ group will no longer be in the main discussion. This would easier the discussion to follow.

Figure 7: Please use the same colors for each period in (a) and (b).

We will keep the same colors for the different periods in (a) and (b).

L. 406: I don't agree with the sentence " The projected ATL3-rainfall signal in the GC+- group is … hardly robust over the Guinea Coast". I see a very robust decreas of rainfall over the green box in figure 10a.

There is a confusion due to incomplete information provided in the submitted mansucript. We were not talking about the changes of the rainfall pattern in the GC+- group, which is a robust decrease as you stated. Rather, we were talking about the rainfall pattern associated with the Atlantic Niño over the 2070-2099 period, as displayed in Fig. A9. We will refer to this map in our revised manuscript.

Supplementary material:

Figure A6 is not discussed in the text

We will remove this useless figure in the revised manuscript. Thank you for this remark.

Technical corrections:

L. 23: Please replace "has moved" by "moves".

This section will be rewritten, to clarify some points highlighted by the reviewer 1. We will take your correction into account. Thank you.

L. 45: Losada et al. 2009 should be Losada et al. 2010a.

Thank you for this remark, it will be taken into account in our revision.

L. 143 to 145: I find this description of the sign-dependent average a bit confusing.

We will improve the description of the sign-dependent average in the revised manuscript.

L. 248: Change "Mohino and Losada (2015)" by "Mohino and Losada (2015), among others".

We will add "among others" to the sentence.

L. 355: Pleas move the sentence " This indicates a weakening… in the eastern equatorial Atlantic" to the end of the paragraph.

We will move the sentence to the end of the paragraph as suggested, thank you.

**RC5**: 'Comment on esd-2021-46', Anonymous Referee #5, 01 Oct 2021

Revision of "Weakened impact of the Atlantic Niño on the future equatorial Atlantic and Guinean Coast rainfall" by K. Worou, H. Goosse, T. Fichefet and F. Kucharski

General comments:

The article is a detailed analysis of the rainfall over the Guinean coast and the relation with Atlantic Niño using a pool of CMIP6 simulations in the historical period compared with observed data. The analysis is relevant, and the article convincingly shows that the rainfall will decrease over the Guinean coast as the Atlantic Niño variability will also decrease in the future.

The paper addresses relevant scientific questions and the conclusions are important for the climate science community.

I have, however, some suggestions for improving the paper from my point of view. I recommend shortening the article; I found a very large document with unnecessary figures. It is, however, good to see a lot of more information in the additional material. I have some ideas of how to reduce it below together with some typos and minor comments

specific comments and technical corrections:

- Typo in line 10 (and more). Please be aware that Bjerknes feedback is referred to Jacob Bjerknes, use Bjerknes feedback instead of Bjerkness feedback.

  Thank you for this correction, we will apply it everywhere in the revised manuscript.

- Line 31 (and also the abstract), You referred to the Atlantic Niño as ATL3, but the community usually defines ATL3 as an index (SST averaged over 20W-0, 3S-3N, Zebiak 1993). Please define the index first and then explain why you identify the Atlantic Niño with ATL3 (you could also use a different acronym).

  We will no longer use ATL3 for the Atlantic Niño mode. We will instead use the Atlantic Equatorial mode acronym (AEM) instead.

- Line 42, a reference should be added.

  This section will be rewritten in the revised manuscript, following the comments of the reviewer 1. We will add correctly references to support our statements.

- Table 2. It would be good to see the boxes within the map (for instance in figure 2)

  These boxes will be added to the figure 2 in the revised manuscript.

- Figure 3b and d, why ERA5 is used as a reference for the std when it is clearly biased in relation with the other observed SST datasets?, please explain in the text or in the figure caption.

  We wanted to be consistent with one reference through our analyses, that is why we chose ERA5 as the reference in the Figure 3b and d. In the revised version of the paper, following the suggestion of the reviewer 1, we will keep only ERA5 to evaluate models performance.

Additionally, we will remove the question of seasonality, so the Figure 3 will be added to the supplementary material.

- Line 132. Why do you remove the quadratic trend instead of the linear trend? It is quite clear the linear trend in the Atl3 SST in the historical period. Please show the trend of the rainfall and SST for the indexes to understand your choice.

[Figure]

*Figure R 9 SST indices of the Atlantic Niño: JAS mean of monthly SST anomalies averaged over the Atlantic Niño area (green curves), for the 1985-2014 period. The linear (blue curves) and quadratic (orange curves) trends are superimposed on each SST index. SST outputs from CMIP6 historical simulations (30 GCMs) and the ERA5 reanalysis are considered.*

The linear in the JAS ATL3 SST is clear in ERA5, whereas in the CMIP6 historical period, it is roughly linear (Fig. R9). The apposition of both linear and quadratic trends in each SST index of the different models show cases where the quadratic trend departs from the linear trend (e.g. GFDL-ESM4). This motivated us to remove the quadratic trend from our data. However, in the revised manuscript, we will remove the linear trend instead of the quadratic trend, as the residuals from the detrended SST indices are similar in both cases (Fig. R10).

[Figure]

*Figure R 10 Residuals of the JAS ATL3 SST index after removing the linear trend (blue curves) and the quadratic trend (orange curves). The indices are computed from 30 CMIP6 data and ERA5 for the 1985-2014 period.*

The analysis of the rainfall time series of the Guinea Coast also shows that in most of the cases, the rainfall trends can be considered as linear (Figure R11). In the case of CESM2-WACCM for example, the two trends are slightly different. However, the residuals from the detrended time series do not exhibit a strong difference in the interannual variability (Figure R12). Therefore, we will consider the trend as linear in the revised manuscript.

[Figure]

*Figure R 11  JAS rainfall index of the Guinea-Coast over the 1985-2014 period. Linear and quadratic trends (blue and orange curves respectively) are superimposed on each rainfall index. Indices are computed from 30 CMIP6 models and ERA5*

[Figure]

*Figure R 12 Residuals of the linearly (blue curves) and quadratically (orange curves) detrended JAS Guinean Coast rainfall indices. Indices are computed from 30 CMIP6 models and the reanalysis ERA5 for the 1985-2014 period.*

- Figure 4a and line 255. From figure 3b and the observations, it is clear the main season for Atl3 would be JJA, why do you decide to compare the observed and simulated Atlantic Niño in JAS? It would be useful to show correlation between SST and rainfall indexes for different seasons to realize which of the seasons is more realistic (maybe in the observation the maximum correlation is between Atl3 SST index in JJA and precipitation GG in JAS).

[Figure]

*Figure R 13 Monthly stratified rainfall anomalies regressed onto the standardized JJA (a) and (JAS) ATL3 index in 30 CMIP6 models and in ERA5 over the 1985-2014 period. The vertical gray band shows the considered season of the ATL3 index.*

The cross-correlation between the JJA and JAS ATL3 index and the monthly rainfall indices of the Guinea Coast indicate a strong ATL3-rainfall covariability during July, August and September in the CMIP6 models (Fig. R13). This result indicates that the amplitude of the rainfall response in the models is maximum in JAS, with a similar order of magnitude, whether the ATL3 index is computed over the JJA or the JAS season. As we are interested in the impact of the Atlantic Niño mode on the rainfall in Guinea-Coast in the CMIP6 models, we will consider the JAS season.

We also note that in the two cases (i.e. JJA and JAS ATL3 index), the CMIP6 ensemble mean rainfall response in June is weak compared to ERA5. Moreover, in ERA5, the rainfall responses to the JJA and JAS Atlantic Niño indices are quite similar and stronger during JJAS (June to September).

- As you exposed in the introduction, deconstructive interaction of Atlantic Niño and ENSO events onto the WA rainfall in some time-periods can conduct into a dipole or monopole of the rainfall anomalies (Losada et al 2012). It would be nice to see how many of those simulated and observed Atl3 indexes are correlated with Niño3 index. Thus, the rainfall pattern in figure 5 could be a mix between local and remote SST drivers.

[Figure]

*Figure R 14 Monthly stratified Niño3 index regressed onto the standardized JAS ATL3 index for different periods. The 1985-2014 period is considered for ERA5 (black curve). The other curves correspond to the ensemble mean response of 30 CMIP6 models over four different periods.*

Thank you for this suggestion. We regressed the monthly Niño3 index onto the standardized JAS Atlantic Niño index for 30 CMIP6 models and ERA5 (Fig. R14). In ERA5, the JAS ATL3 index is negatively correlated with the Niño3 index from April to December. This opposite phase relationship is stronger in November and December. In the January to March, an in-phase relationship is observed. The CMIP6 models ensemble mean response shows an anticorrelation between the JAS ATL3 index and the Niño3 index for all the months during 1985-2014. Thus, the effect of SST in both basins would lead to rainfall anomalies of the same sign over Guinea Coast. In the future, a general decrease of the Niño3-ATL3 relationship is obtained in the 2015-2039 and 2040-2069 period. However, the sign of the correlation between both indices is reversed in the long-term period (2070-2099). This means that in the 2070-2099 period, two rainfall anomalies with opposite signs will interact over Guinea Coast during, and this would reduce the rainfall amplitude associated with the Atlantic Niño.

For the 1985-2014, 2015-2039, 2040-2069 and 2070-2099 periods, there are, respectively, 9, 7, 9 and 7 models which show a significant correlation between the JAS Niño3 and ATL3 indices (Fig. R15). This correlation is not significant for ERA5 during JAS. These results will be added to the revised manuscript.

[Figure]

*Figure R 15 JAS Niño3 index correlation with the JAS ATL3 index over four different periods. 30 CMIP6 models and the reanalysis ERA5 are analyzed. Significant regression coefficients at 90% confidence level (Student test) are highlighted with a black box.*

- Line 264. Stronger correlation between Atl3-SST and SSH in models than in ORAS5 implies stronger Bjerknes feedback in the models, which I did not expected from CMIP5 models analysis (for instance Dippe et al 2018, DOI 10.1007/s00382-017-3943-z). It would be nice to see surface wind superimposed on figures 6 u-x to illustrate the 3 elements of the Bjerknes feedback.

  The 850 hPa horizontal wind will be added to the SSH maps in the Figure (6), and we will also refer to Dippe et al 2018, to compare the strength of the Bjerknes feedback in CMIP5 and CMIP6 models.

- Paragraph from line 280 please reduce or suppress, I do not see that OC+ models are explaining important differences from GC+ models. Remove (or reduce) and explain later in the text (beginning of section 5.2)

The text from the lines 280 to 305 will be suppressed, and the OC+ group will be removed from the main discussions.

- Figure 8 is a very interesting and illustrative view of the processes and the trends, however, Figure 9 is not necessary (figure 9a is certainly illustrative of where the mean change occurs but figure 9 overall is redundant). From my point of view figure 9 should be removed or put in the additional material. Conclusions on this part could be explained with figure 8 alone.

  Figure 9 will be put in the additional material, thank you for the suggestion.

- Line 380 should start a new section 5.2

  The section 5.2 will start from the line 380 as suggested.

- Line 397 typo zonal

  Thank you for this correction.

- I found all the discussion about figure 10 of OC models very unnecessary, indeed in your abstract you don't mention such differences. The main result about this in the abstract is "higher confidence in the reduction of the rainfall associated with atl4 over the Atlantic Ocean than over the Guinea coast". It is appreciated the detailed analysis of the different model flavours but It doesn't give any light into the main conclusion. I will leave the GG models alone, and the OC models in the additional material. Also, figure 11 b is not necessary for the conclusions, I would go for 11a alone. Please enlarge Figure 11a.

  We agree with the comment. We will remove the OC categories from the Figure 10 and we will put them into the supplementary material.

  The reviewer 1 suggests removing the discussion about the extension of the Sahara heat low to the tropical north Atlantic. We will then move the enlarged Fig. (11a) to the supplementary material, and the Fig. (11b) will be suppressed.

- Line 471 remove more

  Thank you for the correction. It will be taken into account.